# ESOTERIC LANGUAGE MODELS

## ABSTRACT

Diffusion-based language models offer a compelling alternative to autoregressive (AR) models by enabling parallel and controllable generation. Among this family of models, Masked Diffusion Models (MDMs) achieve the strongest performance but still underperform AR models in perplexity and lack key inference-time efficiency features—most notably, KV caching. In this work, we introduce Eso-LMs, a new family of models that **fuses AR and MDM paradigms**, enabling smooth interpolation between their perplexities while overcoming their respective limitations. Crucially, we **introduce KV caching for MDMs** while preserving parallel generation, significantly improving inference efficiency. Combined with an optimized sampling schedule, our method achieves a new state of the art on the speed-quality Pareto frontier for unconditional generation. On long contexts, our method achieves $14 - 65\times$ faster inference than standard MDMs and $3 - 4\times$ faster inference than prior semi-autoregressive approaches.

## 1 INTRODUCTION

A paradigm shift is underway in language modeling: autoregressive (AR) language models, long considered the gold standard, are now being rivaled by diffusion language models for standard language generation. Recent works (Sahoo et al., 2024a; Schiff et al., 2025) show that Masked Diffusion Models (MDMs) are closing the gap with AR models on small-scale language benchmarks, and even outperform them on tasks involving discrete structures, such as molecular generation (Schiff et al., 2024; Lee et al., 2025) and graph generation (Liu et al., 2023). When scaled to larger sizes (e.g., 8B parameters), MDMs match models like LLaMA on challenging datasets in math, science, and tasks such as reverse poem completion (Nie et al., 2025).

These results make MDMs a compelling alternative to AR models. However, they suffer from two key limitations: (1) **Inference speed**: Despite supporting parallel generation, MDMs are significantly slower than AR models in practice, largely due to the lack of KV caching, which is a crucial optimization for real-time applications like chat systems. (2) **Generation quality**: MDMs still show a noticeable likelihood gap on more complex language modeling tasks (Sahoo et al., 2024a).

Recently proposed BD3-LMs (Arriola et al., 2025) address the speed issue by introducing a semi-autoregressive generation strategy. These models perform diffusion over fixed-length blocks of text sequentially. Because previously denoised blocks can be cached, BD3-LMs partially support KV caching and are faster than standard MDMs. However, we identify two key shortcomings in BD3-LMs: (1) **Degraded samples at low sampling steps**: When the number of denoising steps is reduced for faster inference, BD3-LMs exhibit severe degradation in sample quality and diversity—-worse than both AR (at high Number of Function Evaluations (NFEs), i.e., neural network forward passes) and other diffusion models (at low NFEs) (Sec. A.1 and Sec. 5.2). (2) **Incomplete caching**: While KV caching is possible across blocks, intra-block diffusion still lacks KV support, limiting overall speed gains.

To address these challenges, we propose a new language modeling paradigm that fuses autoregressive and masked diffusion approaches. Our model is trained with a hybrid loss—a combination of AR and MDM objectives—which allows it to interpolate smoothly between the two paradigms in terms of perplexity. This requires two key innovations: (1) A revised attention mechanism in the denoising transformer to support both AR and MDM styles of generation. (2) A new training and sampling procedure that enables KV caching within the diffusion phase, a feature previously unavailable in MDMs. Due to the unconventional nature of this hybrid design, we name our method **Eso**teric **L**anguage **M**odels (Eso-LMs).

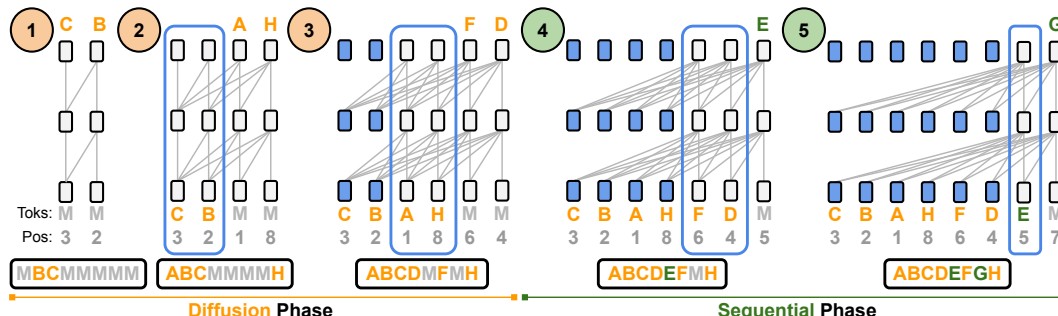

Figure 1: Efficient generation of an example sequence with our proposed Eso-LMs. During **Diffusion** Phase, Eso-LMs denoise one or more, potentially non-neighboring mask tokens ($\mathbb{M}$) per step. During **Sequential** Phase, Eso-LMs denoise the remaining mask tokens one at a time from left to right. Eso-LMs allow for **KV caching in both phases** using just **a single unified KV cache**: **blue** bounding boxes enclose transformer cells that are building their KV cache; a cell becomes **blue** once its KV cache is built. The sequences below the transformers depict tokens in their natural order.

In summary, our contributions are as follows. (1) We propose a new hybrid framework for language modeling: one that fuses AR and MDM paradigms and outperforms the previous hybrid approach, BD3-LMs. (2) We show that our proposed Eso-LMs achieve fine-grained interpolation between AR and MDM perplexities, narrowing the gap to AR models (Sec. 5.1). (3) By enabling KV caching during diffusion while preserving parallel generation, Eso-LMs achieve a new state of the art on the speed-quality Pareto frontier for unconditional generation. BD3-LMs degrade at low sampling steps, while our method remains competitive with MDMs in the low NFE regime and with AR in the high NFE regime (Sec. 5.2). (4) At long contexts, Eso-LMs provide $14 - 65\times$ faster inference than standard MDMs and $3 - 4\times$ faster inference than KV-cached semi-autoregressive baselines (Sec. 5.3).

## 2 BACKGROUND

**Notation** We represent scalar discrete random variables that can take $K$ values as 'one-hot' column vectors and define $\mathcal{V} \in \{\mathbf{x} \in \{0,1\}^K : \sum_{i=1}^K \mathbf{x}_i = 1\}$ as the set of all such vectors. In the context of language modeling, $K$ is the vocabulary size and $\mathcal{V}$ is the vocabulary. Let $\mathbf{m} \in \mathcal{V}$ be a special mask vector such that its $K$-th entry is one, i.e., $\mathbf{m}_K = 1$. Define $\mathrm{Cat}(\cdot; \boldsymbol{\pi})$ as the categorical distribution over $K$ classes with probabilities given by $\boldsymbol{\pi} \in \Delta^K$, where $\Delta^K$ denotes the $K$-simplex. Additionally, let $\langle \mathbf{a}, \mathbf{b} \rangle$ denote the dot product between vectors $\mathbf{a}$ and $\mathbf{b}$. We use parentheses ( ) to denote ordered sets (tuples) and curly brackets $\{\}$ to denote unordered sets. $|A|$ denotes the cardinality of the set $A$.

MDMs feature two salient orderings: the sequence order and the denoising order. We use a permutation $\sigma$ to describe the relationship between these orderings. Let $\mathcal{P}_L$ denote the set of all permutations of $[L] = \{1, \ldots, L\}$. $\sigma \in \mathcal{P}_L$ is an ordered set (tuple) and also serves as a bijective function: $\sigma(l)$ is the position in sequence order that appears $l^{\text{th}}$ in denoising order $\sigma$, and $\sigma^{-1}(i)$ is the position in denoising order for the $i^{\text{th}}$ position in sequence order. For example, $\sigma = (2, 4, 1, 3)$ denotes a denoising order of $(1, 2, 3, 4)$; $\sigma^{-1}(4) = 2$ means the $4^{\text{th}}$ token in sequence is the $2^{\text{nd}}$ one to denoise.

Let $\mathbf{x} \in \mathcal{V}^L$ denote a sequence of length $L$ with no mask tokens, and let $\mathbf{x}^\ell$ denote the $\ell^{\text{th}}$ entry in $\mathbf{x}$. Note that $\mathbf{x}^l$ is one-hot under our notation. We use the term 'token index' to refer to the position of a token in the original ordering, e.g., the token index for $\mathbf{x}^l$ is $l$. Let $(\mathbf{z}_t)_{t \in [0,1]} \in \mathcal{V}^L$ denote a sequence of length $L$ that may contain mask tokens. Let $\mathcal{M}(\mathbf{z}_t) = \{\ell \mid \mathbf{z}_t^\ell = \mathbf{m}\}$ denote token indices of mask tokens in $\mathbf{z}_t$ and $\mathcal{C}(\mathbf{z}_t) = \{\ell \mid \mathbf{z}_t^\ell \neq \mathbf{m}\}$ denote token indices of clean tokens in $\mathbf{z}_t$.

Let $\oplus : \mathcal{V}^m \times \mathcal{V}^n \to \mathcal{V}^{m+n}$ denote a concatenation operator on two sequences $\mathbf{x} = (\mathbf{x}^1, \mathbf{x}^2, \ldots, \mathbf{x}^m)$ and $\mathbf{z} = (\mathbf{z}^1, \mathbf{z}^2, \ldots, \mathbf{z}^n)$ of length $m$ and $n$. When the concatenated sequence $\mathbf{x} \oplus \mathbf{z}$ is fed into the transformer, $\mathbf{x}$ and $\mathbf{z}$ carry the same positional embeddings as they would if they were fed into a transformer independently. Let $\odot : \mathcal{V}^m \times \mathcal{V}^n \to \mathcal{V}^m$ denote a substitution operator; for any $\mathbf{z} \in \mathcal{V}^m$ and $\mathbf{x} \in \mathcal{V}^n$ with $m > n$, the output $\mathbf{y} = \mathbf{z} \odot \mathbf{x}$ is given by: $\mathbf{y}^{1:n} = \mathbf{x}$ and $\mathbf{y}^{n+1:m} = \mathbf{z}^{n+1:m}$.

## 2.1 AUTOREGRESSIVE MODELS

Given a sequence $\mathbf{x} \in \mathcal{V}^L \sim q_{\text{data}}$, AR models define the following factorization of the joint distribution: $\log p_\theta(\mathbf{x}) = \sum_{\ell=1}^{L} \log p_\theta(\mathbf{x}^\ell \mid \mathbf{x}^{<\ell})$, where the model $p_\theta$ is usually parameterized by a causal transformer (Vaswani et al., 2017) model. Sampling takes $L$ steps or NFEs but each is computationally efficient due to KV caching. AR models achieve the best likelihood and generation quality.

## 2.2 MASKED DIFFUSION MODELS

Diffusion models learn to reverse a forward corruption process $q$, which transforms clean data $\mathbf{x} \sim q_{\text{data}}$ in $\mathcal{V}^L$ into a sequence of latent variables $\mathbf{z}_t$ for $t \in [0, 1]$, each representing an increasingly noisy version of $\mathbf{x}$ (Ho et al., 2020; Sohl-Dickstein et al., 2015; Song & Ermon, 2019). In MDMs (Sahoo et al., 2024a; Shi et al., 2025; Ou et al., 2025), the forward masking process $q$ factors independently across the sequence $\mathbf{x}$, i.e., $q_t(.|\mathbf{x}) = \Pi_\ell q_t(.|\mathbf{x}^\ell)$ and each token $\mathbf{x}^l$ is progressively interpolated with a fixed target distribution $\text{Cat}(.; \mathbf{m})$. The marginal of $\mathbf{z}_t^\ell \sim q_t(.|\mathbf{x}^\ell)$ at time $t$ is given by:

$$q_t(.|\mathbf{x}^\ell) = \text{Cat}(.; \alpha_t \mathbf{x}^\ell + (1 - \alpha_t)\mathbf{m}), \tag{1}$$

where $\alpha_t \in [0, 1]$ is a strictly decreasing function in $t$ with $\alpha_0 \approx 1$ and $\alpha_1 \approx 0$. Sahoo et al. (2024a) show that the reverse posterior $q_{s|t}(.|\mathbf{z}_t^\ell, \mathbf{x}^\ell)$ over $\mathbf{z}_s^\ell$ for $s < t$ is given by

$$q_{s|t}(.|\mathbf{z}_t^\ell, \mathbf{x}^\ell) = \begin{cases} \text{Cat}(.; \mathbf{z}_t^\ell) & \mathbf{z}_t^\ell \neq \mathbf{m}, \\ \text{Cat}\left(.; \frac{(1-\alpha_s)\mathbf{m}+(\alpha_s-\alpha_t)\mathbf{x}^\ell}{1-\alpha_t}\right) & \mathbf{z}_t^\ell = \mathbf{m}. \end{cases} \tag{2}$$

Given a denoising model $\mathbf{x}_\theta : \mathcal{V}^L \rightarrow (\Delta^K)^L$, the reverse unmasking process $p_\theta(.|\mathbf{z}_t)_{s|t}$ over the sequence $\mathbf{z}_s$ is parameterized by

$$p_\theta(.|\mathbf{z}_t)_{s|t} = \prod_\ell^L p_\theta^\ell(.|\mathbf{z}_t)_{s|t} = \prod_\ell^L q_{s|t}^\ell(.|\mathbf{z}_t^\ell, \mathbf{x}^\ell = \mathbf{x}_\theta^\ell(\mathbf{z}_t)). \tag{3}$$

Sahoo et al. (2024a); Shi et al. (2025); Ou et al. (2025) show that Negative Evidence Lower Bound (NELBO) for this method is

$$\mathcal{L}_{\text{MDM}}(\mathbf{x}) = \mathbb{E}_{q, t \sim [0,1]} \left[ \frac{\alpha_t'}{1 - \alpha_t} \sum_{\ell \in \mathcal{M}(\mathbf{z}_t)} \log \langle \mathbf{x}_\theta^\ell(\mathbf{z}_t), \mathbf{x}^\ell \rangle \right], \tag{4}$$

which is a weighted average of masked language modeling losses (Devlin et al., 2018) computed only on the masked positions $\mathcal{M}(\mathbf{z}_t)$.

To generate a sequence of length $L$, the reverse diffusion process starts from a fully masked sequence $\mathbf{z}_{t=1}$, where $\mathbf{z}_{t=1}^\ell = \mathbf{m}$ for $\ell = 1, \ldots, L$. It proceeds for $T$ steps, with each $\mathbf{z}_s^\ell$ independently sampled from $p_\theta(.|\mathbf{z}_t)_{s|t}$ as defined in (3). More concretely, the sampler iterates from $t = 1$ to $1 - dt$ (inclusive) for $T$ steps with $dt = 1/T$ and $s = t - dt$. Each step consists of two sub-steps. First, the number of mask positions to denoise is sampled using

$$n_t \sim \text{Binom}\left(n = n_t^{\text{remaining}}, p = \frac{\alpha_s - \alpha_t}{1 - \alpha_t}\right), \tag{5}$$

where $n_t^{\text{remaining}} = L - \sum_{t'>t} n_{t'}$ is the number of remaining mask positions. Second, $n_t$ positions are randomly selected from the remaining mask positions and are independently denoised according to probabilities provided by $\mathbf{x}_\theta(\mathbf{z}_t)$. Once a position is unmasked, it remains fixed. Since multiple tokens can be denoised in parallel per step, the total number of steps or NFEs can be less than $L$, enabling faster generation. However, each forward pass is computationally expensive due to applying the bidirectional transformer in $\mathbf{x}_\theta(\mathbf{z}_t)$ over the entire context length.

## 2.3 BLOCK DISCRETE DIFFUSION MODELS

Block Denoising Diffusion Discrete Language Models (BD3-LMs) (Arriola et al., 2025) autoregressively model blocks of tokens and perform masked diffusion modeling (Sec. 2.2) within each block. By changing the size of blocks, BD3-LMs interpolate AR models and MDMs. BD3-LMs

group tokens in $\mathbf{x}$ into $B$ blocks of $L'$ consecutive tokens with $B = L/L'$, where $B$ is an integer. The likelihood over $\mathbf{x}$ factorizes autoregressively over blocks as $-\log p_\theta(\mathbf{x}) = -\sum_{b=1}^{B} \log p_\theta(\mathbf{x}^b \mid \mathbf{x}^{<b}) \leq \sum_{b=1}^{B} \mathcal{L}_{\text{MDM}}(\mathbf{x}^b; \mathbf{x}^{<b})$, where $p_\theta(\mathbf{x}^b \mid \mathbf{x}^{<b})$ is a conditional MDM and $\mathcal{L}_{\text{MDM}}(\mathbf{x}^b; \mathbf{x}^{<b})$ is the NELBO for MDLM as defined in (4), applied sequentially across blocks. During generation, we use $T' = T/L'$ to denote the number of diffusion sampling steps per block.

# 3 ESOTERIC LANGUAGE MODELS

In this section, we propose a new paradigm for language modeling: **Eso**teric **L**anguage **M**odels (Eso-LMs), which form a symbiotic combination of AR models and MDMs.

While AR models currently lead in language modeling performance, they generate tokens sequentially, making them slow at inference. In contrast, MDMs generate multiple tokens in parallel and are more controllable (Schiff et al., 2025; Nisonoff et al., 2024), but they typically yield higher perplexity than AR models (Sahoo et al., 2024a; 2025). Can we combine their strengths? In response, we introduce a hybrid approach where some tokens are generated in parallel via MDMs and the rest sequentially in a left-to-right fashion. This raises two key questions. (1) Can we compute the likelihood of such a generative model? We address this question by showing that Eso-LMs admit a principled bound on the true likelihood. (2) How can we adapt the attention mechanism so that a single transformer (Vaswani et al., 2017) can support both styles of generation? We address this question in Sec. 4.

## 3.1 FUSING AUTOREGRESSIVE MODELS AND MASKED DIFFUSION

Let $\mathbf{x} \in \mathcal{V}^L \sim q_{\text{data}}(\mathbf{x})$ be a sample from the data distribution, and let $p_\theta$ be our model distribution parameterized by $\theta$. Eso-LMs decompose $p_\theta$ into two components: an AR model $p_\theta^{\text{AR}}$ and an MDM $p_\theta^{\text{MDM}}$. The MDM generates a partially masked sequence $\mathbf{z}_0 \in \mathcal{V}^L \sim p_\theta^{\text{MDM}}(\mathbf{z}_0)$, and the AR model finishes the remaining unmasking steps in an auto-regressive left-to-right manner: $p_\theta^{\text{AR}}(\mathbf{x}|\mathbf{z}_0)$. The marginal likelihood of such a hybrid generative process is:

$$p_\theta(\mathbf{x}) = \sum_{\mathbf{z}_0 \in \mathcal{V}^L} p_\theta^{\text{AR}}(\mathbf{x}|\mathbf{z}_0) p_\theta^{\text{MDM}}(\mathbf{z}_0). \tag{6}$$

Although this sum is intractable, we can compute a variational bound on the true likelihood using a posterior $q(\mathbf{z}_0|\mathbf{x})$ (Kingma & Welling, 2014). Since $p_\theta^{\text{MDM}}$ models masked sequences, we choose $q$ to be a simple masking distribution. Specifically, we set $q$ to $q_0(\mathbf{z}_0|\mathbf{x})$ as defined in (1), which independently masks each token $(\mathbf{x}^\ell)_{\ell \in [L]}$ with probability $1 - \alpha_0$, where $\alpha_0 \in [0, 1]$; intuitively, $\alpha_0$ is the expected fraction of clean tokens in $\mathbf{x}$ by MDM. This leads to the following variational bound:

$$-\log p_\theta(\mathbf{x}) \leq = -\mathbb{E}_{\mathbf{z}_0 \sim q_0(.|\mathbf{x})} \left[ \sum_{\ell \in \mathcal{M}(\mathbf{z}_0)} \log p_\theta^{\text{AR}}(\mathbf{x}^\ell|\mathbf{z}_0, \mathbf{x}^{<\ell}) \right] + \mathrm{D}_{\text{KL}}(q_0(\mathbf{z}_0|\mathbf{x}) \| p_\theta^{\text{MDM}}(\mathbf{z}_0)). \tag{7}$$

Inside the expectation is the joint AR likelihood over masked positions $\ell \in \mathcal{M}(\mathbf{z}_0)$, conditioned on clean tokens in $\mathbf{z}_0$. For AR, the denoising network $\mathbf{x}_\theta : \mathcal{V}^L \to (\Delta^K)^L$ operates on the input $\mathbf{z}_0 \odot \mathbf{x}^{<\ell}$, where the substitution operator $\odot$ replaces the first $l - 1$ tokens in $\mathbf{z}_0$ with $\mathbf{x}^{<\ell}$. For each $\ell \in \mathcal{M}(\mathbf{z}_0)$, $\mathbf{x}_\theta^\ell(\mathbf{z}_0 \odot \mathbf{x}^{<\ell})$ approximates the distribution of the clean token $\mathbf{x}^\ell$ given $\mathbf{x}^{<\ell}$ and $\mathbf{z}_0$, which may include clean tokens beyond position $\ell$. In Suppl. B.1, we analyze the KL term and show that the NELBO is:

$$\mathcal{L}_{\text{NELBO}}(\mathbf{x}) = \mathbb{E}_{\mathbf{z}_0 \sim q_0} \underbrace{\left[ - \sum_{\ell \in \mathcal{M}(\mathbf{z}_0)} \log \langle \mathbf{x}_\theta^\ell(\mathbf{z}_0 \odot \mathbf{x}^{<\ell}), \mathbf{x}^\ell \rangle \right]}_{\text{AR loss}} + \underbrace{\int_{t=0}^{t=1} \frac{\alpha_t'}{1 - \alpha_t} \mathbb{E}_{\mathbf{z}_t \sim q_t} \left[ \sum_{\ell \in \mathcal{M}(\mathbf{z}_t)} \log \langle \mathbf{x}_\theta^\ell(\mathbf{z}_t), \mathbf{x}^\ell \rangle \right] dt}_{\text{MDM loss}}, \tag{8}$$

where we set the diffusion noise schedule $\alpha_t$ to be the standard log-linear schedule $\alpha_t = \alpha_0(1 - t)$.

**Interpolating between AR and Diffusion** When $\alpha_0 = 1$, the posterior sample $\mathbf{z}_0 = \mathbf{x}$, and all tokens are generated by the MDM; hence, the AR loss is zero in (8), and $\mathcal{L}_{\text{NELBO}}$ reduces to the MDM loss. Conversely, when $\alpha_0 = 0$, all tokens are masks in $\mathbf{z}_0$, and the MDM loss vanishes, reducing $\mathcal{L}_{\text{NELBO}}$ to the AR loss. Thus, Eso-LMs interpolate between AR and MDM paradigms, controlled by the hyperparameter $\alpha_0$.

## 3.2 SAMPLING

We use the two-stage sampling procedure from (6). To draw $\mathbf{x}$, we first sample a partially masked sequence $\mathbf{z}_0 \sim p_\theta^{\text{MDM}}$ and then denoise the remaining mask tokens left-to-right with $p_\theta^{\text{AR}}$.

**Denoising Schedule** During sampling, we pre-compute the order in which tokens will be denoised under the standard ancestral sampler. We denote the *diffusion denoising schedule* by $\mathcal{S}^{\text{MDM}} = (S_1, \ldots, S_{1/T})$, where $S_t$ is a tuple of mask token indices denoised at diffusion step $t$, and $T$ is the total number of denoising steps. As with MDLM, the number of mask tokens to denoise per step is sampled using (5) and their indices $S_t$ are randomly selected among the remaining mask positions. However, MDLM uses the noise schedule $\alpha_t = 1 - t$ while Eso-LMs use $\alpha_t = \alpha_0(1 - t)$, yielding an expected fraction of clean tokens of $\alpha_{t=0} = \alpha_0$ after diffusion (1), which could be less than 1. We define the *AR denoising schedule* as $\mathcal{S}^{\text{AR}} = ((i) \mid i \in \mathcal{M}(\mathbf{z}_0))$, where the mask indices $\mathcal{M}(\mathbf{z}_0)$ appear in strictly ascending order. The *unified denoising schedule* is then given by $\mathcal{S} = \mathcal{S}^{\text{MDM}} \cup \mathcal{S}^{\text{AR}}$, which concatenates the two schedules to partition $[L]$. When $\alpha_0 = 1$, all tokens are generated by diffusion so $\mathcal{S} = \mathcal{S}^{\text{MDM}}$ and $\mathcal{S}^{\text{AR}} = \varnothing$; when $\alpha_0 = 0$, all tokens are generated sequentially so $\mathcal{S} = \mathcal{S}^{\text{AR}}$ and $\mathcal{S}^{\text{MDM}} = \varnothing$. See Alg. 1 for the concise algorithm for pre-computing the unified denoising schedule and Suppl. B.3 for an illustrative example.

---

**Algorithm 1** Pre-computing Denoising Schedule

**Input:** sequence length $L$, expected fraction of tokens by diffusion $\alpha_0$, diffusion steps $T$

$n^{\text{remaining}} \leftarrow L$, $\mathrm{d}t \leftarrow \frac{1}{T}$
$\mathcal{S}^{\text{MDM}} \leftarrow ()$, $\mathcal{S}^{\text{AR}} \leftarrow ()$
$\mathcal{M} = \{1, \ldots, L\}$ $\qquad \triangleright$ Set of all mask tokens
// Diffusion Denoising Schedule
**for** $t \in [1, 1 - \mathrm{d}t, \ldots, 2\mathrm{d}t, \mathrm{d}t]$ **do**
$\quad \alpha_t \leftarrow \alpha_0(1 - t)$
$\quad \alpha_s \leftarrow \alpha_0(1 - t + \mathrm{d}t)$
$\quad n_t \sim \text{Binom}(n = n^{\text{remaining}}, p = \frac{\alpha_s - \alpha_t}{1 - \alpha_t})$
$\quad S_t \leftarrow \text{SampleWithoutReplace}(\mathcal{M}, n_t)$
$\quad \mathcal{S}^{\text{MDM}} \leftarrow \mathcal{S}^{\text{MDM}} \cup (S_t)$
$\quad \mathcal{M} \leftarrow \mathcal{M} - S_t$
$\quad n^{\text{remaining}} \leftarrow n^{\text{remaining}} - n_t$
**end for**
// Autoregressive Denoising Schedule
**for** $i \in \mathcal{M}$ **do**
$\quad \mathcal{S}^{\text{AR}} \leftarrow \mathcal{S}^{\text{AR}} \cup ((i))$
**end for**
**return** $\mathcal{S}^{\text{MDM}} \cup \mathcal{S}^{\text{AR}}$

---

One of our goals is to eliminate redundancies at inference time in MDMs. Recall that sampling begins with a fully masked sequence $\mathbf{z}_{t=1} = \mathbf{m}^{1:L}$. Standard ancestral sampling as implemented in MDLM (Sec. 2.2) updates only a subset of mask tokens at each step, but still performs a forward pass over the full sequence, wasting FLOPs. To improve efficiency, we restrict the forward pass at step $k$ to only the previously denoised tokens and the current mask tokens to be updated, i.e., $\cup_{i \leq k} S_i$. This substantially reduces computation, especially for long sequences. Building on this sampling procedure, we will describe a method in Sec. 4.1.1 that replaces bidirectional attention in the denoising transformer with causal attention, unlocking KV caching across diffusion steps.

## 3.3 IMPORTANCE WEIGHTED NELBO

For MDMs, the likelihood measures how well they model the data distribution under infinitesimal diffusion steps, where at most one token is denoised or masked out per step. In this limiting case, MDMs are equivalent to any-order AR models, which has the following importance-weighted bound on the negative log-likelihood (Burda et al., 2015; Shih et al., 2022; Hoogeboom et al., 2021):

$$-\log p(\mathbf{x}) \leq -\mathbb{E}_{\sigma_1, \ldots, \sigma_K} \left[ \log \frac{1}{K} + \log \sum_{k=1}^{K} \exp \left( \sum_{l=1}^{L} \log p_\theta(x_{\sigma_k(l)} \mid x_{\sigma_k(<l)}) \right) \right], \quad (9)$$

where $\sigma$ is the denoising ordering introduced in Sec. 2. This bound is tight as $K \to \infty$. This bound is intractable for MDLM because its evaluation requires $L$ forward passes. In contrast, given some $\mathbf{x}$ and $\sigma$, we can evaluate the $\sigma$-order AR term (inside the exponent) for Eso-LMs in a single forward pass (Sec. 4.1.2). We apply this technique to evaluate Eso-LMs in Sec. 5.1.

## 4 ATTENTION MECHANISMS FOR THE SHARED DENOISING TRANSFORMER

In this section, we introduce a unified attention scheme that supports both sequential (AR) and parallel (MDM) generation using a shared transformer architecture. Our core technical contribution

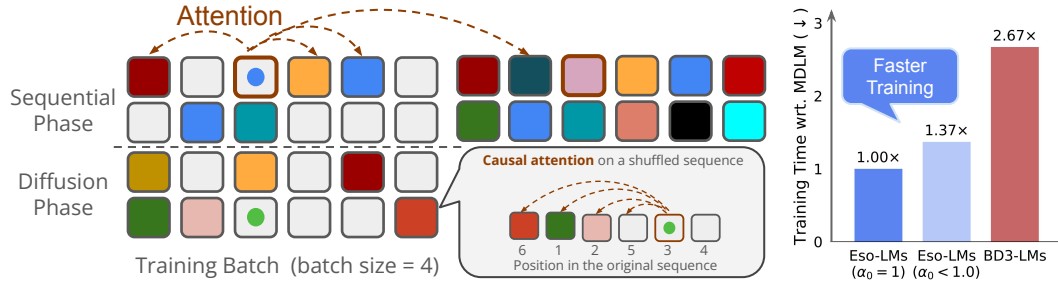

Figure 2: **(Left)** To train a transformer to support both sequential and diffusion generation with KV caching, we use half of the training batch (2 sequences in this example) for diffusion training and the other half for sequential training. Tokens for sequential training are masked with probability $1 - \alpha_0$, while tokens for diffusion training are masked with $p \sim \text{Unif}[1 - \alpha_0, 1]$. (•) For sequential training, a mask token attends to unmasked clean tokens and clean versions of mask tokens on its left. (•) For diffusion training, a mask token attends to all clean tokens and prior mask tokens after shuffling. **(Right)** Eso-LMs have similar training time to MDLM and are much faster to train than BD3-LMs.

is a flexible attention mechanism that reconciles the architectural mismatch between AR models—which require causal attention and shift-by-one decoding—and MDMs—which rely on bidirectional attention. To achieve this, we introduce an attention bias matrix $A \in \{-\infty, 0\}^{L' \times L'}$, where $L'$ is the input length, that modulates the standard attention as: $\text{SELF-ATTENTION}(Q, K, V, A) = \texttt{softmax}\left(\frac{QK^\top}{\sqrt{d}} + A\right)V$ where $Q, K, V \in \mathbb{R}^{L' \times d}$ denote the query, key, and value matrices. Entries of $A$ control information flow: $A_{i,j} = 0$ "permits" and $A_{i,j} = -\infty$ "blocks" attention from token $i$ to $j$.

## 4.1 TRAINING

Our training objective (8) has two components: the AR loss and the diffusion loss. Given a batch of clean sequences, we train a fraction $\kappa$ with the diffusion objective and the remaining $1 - \kappa$ with the AR objective (Fig. 2). We set $\kappa = 0.5$ using an experiment (Sec. 4); for $\alpha_0 = 1$, we set $\kappa = 1$.

### 4.1.1 DIFFUSION PHASE

The diffusion inference scheme (Sec. 3.2) motivates our training setup. We note three properties: (i) clean tokens are generated in random order, (ii) mask tokens are denoised using only clean tokens but clean tokens do not attend to mask tokens, and (iii) bidirectional attention used in MDMs (Austin et al., 2021; Lou et al., 2024; Sahoo et al., 2024a) prevents KV-caching. We propose a simple alternative: given $\mathbf{z}_t \sim q_t(.|\mathbf{x})$, shuffle $\mathbf{z}_t$ with the natural constraint that clean tokens precede masked tokens, and replace bidirectional with causal attention (Fig. 6; more details in Suppl. B.4).

### 4.1.2 SEQUENTIAL PHASE

The AR component of (8) applies a cross-entropy loss on logits for each mask token $(\mathbf{z}_0^i)_{i \in \mathcal{M}(\mathbf{z}_0)}$, requiring its left context to be clean. This is non-trivial because not all mask tokens have a fully clean left context in $\mathbf{z}_0$. We address this by feeding the concatenated sequence $\mathbf{z}_0 \oplus \mathbf{x}$ into the transformer and designing a specialized attention mask so that each $(\mathbf{z}_0^i)$ can also attend to $\mathbf{x}^{<i}$. During sampling, this concatenation is unnecessary. Since only half of each batch is used for sequential training, the doubled sequence length due to concatenation has relatively small impact on training speed (Fig. 2).

**Attention Mask** At inference, KV values for clean tokens in $\mathbf{z}_0$—generated in random order by diffusion—must be reused. Training must therefore enforce causal attention over different random orders among clean tokens $\{\mathbf{x}^i \mid i \in \mathcal{C}(\mathbf{z}_0)\}$ to avoid invalidating the KV cache. We sample a permutation $\sigma \sim \mathcal{P}_L$ such that (i) clean tokens precede mask tokens, and (ii) mask tokens remain in

natural order. The following $2L \times 2L$ attention bias matrix $A$ enforces correct information flow:

$$A_{i,j} = 0 \qquad \text{if } i = j \ \forall (i,j) \in \mathcal{M}(\mathbf{z}_0) \times \mathcal{M}(\mathbf{z}_0) \tag{10}$$
$$A_{i,j+L} = 0 \qquad \forall (i,j) \in \mathcal{M}(\mathbf{z}_0) \times \mathcal{C}(\mathbf{z}_0) \tag{11}$$
$$A_{i,j+L} = 0 \qquad \text{if } i > j \ \forall (i,j) \in \mathcal{M}(\mathbf{z}_0) \times \mathcal{M}(\mathbf{z}_0) \tag{12}$$
$$A_{i+L,j+L} = 0 \quad \text{if } \sigma^{-1}(i) \geq \sigma^{-1}(j) \ \forall (i,j) \in \mathcal{C}(\mathbf{z}_0) \times \mathcal{C}(\mathbf{z}_0) \tag{13}$$
$$A_{i+L,j+L} = 0 \quad \forall (i,j) \in \mathcal{M}(\mathbf{z}_0) \times \mathcal{C}(\mathbf{z}_0) \tag{14}$$
$$A_{i+L,j+L} = 0 \quad \text{if } i \geq j \ \forall \ (i,j) \in \mathcal{M}(\mathbf{z}_0) \times \mathcal{M}(\mathbf{z}_0) \tag{15}$$
$$A_{i,j} = -\infty \qquad \text{otherwise.} \tag{16}$$

Refer Fig. 8 for an illustrative example. This construction ensures: a mask token $(\mathbf{z}_0^i)_{i \in \mathcal{M}(\mathbf{z}_0)}$ attends to (i) itself (10), (ii) the clean tokens in $\mathbf{z}_0$ (equivalently $(\mathbf{x}^i)_{i \in \mathcal{C}(\mathbf{z}_0)}$) (11), and (iii) the clean versions of mask tokens on its left (12). A clean token $(\mathbf{z}_0^i)_{i \in \mathcal{C}(\mathbf{z}_0)}$ can attend to anything because no other token attends to them. Tokens $\{\mathbf{x}^i | i \in \mathcal{C}(\mathbf{z}_0)\}$ have causal attention per $\sigma$ (13). A clean token corresponding to a mask token, $(\mathbf{x}^i)_{i \in \mathcal{M}(\mathbf{z}_0)}$, attends to $\{\mathbf{x}^j | j \in \mathcal{C}(\mathbf{z}_0)\}$ (14) and $\{\mathbf{x}^j | j \in \mathcal{M}(\mathbf{z}_0), i \geq j\}$ (15).

**Simplified Implementation**   When the rows and columns of each of the four $L \times L$ blocks are sorted by $\sigma$, $A$ shows classic attention patterns (Fig. 8) that are simple to code in PyTorch (Fig. 9).

## 4.2   SAMPLING

At each sampling step, we perform a forward pass of clean tokens decoded in the previous step for KV caching and mask tokens corresponding to positions to decode in the current step (Fig. 1). We unlock two features for efficiency: (1) KV caching during diffusion phase and (2) a shared KV cache for diffusion and sequential phases. Also, our sampler can decode according to any denoising schedules, even ones not seen during training, which leads to interesting inference-time trade-offs (Sec. 5.2).

**Attention Mask**   More concretely, during sampling step $k$, given a partially masked sequence $\mathbf{z}_k$, the denoising model is required to denoise the mask tokens $\{\mathbf{z}_k^i | i \in S_k\}$ where $S_k \in \mathcal{S}$. Recall that $\mathcal{S} = (S_1, \ldots, S_K)$ is the unified denoising schedule introduced in Sec. 3.2. We perform a forward pass on the subset of tokens $\{\mathbf{z}_k^i | i \in \mathcal{C}(\mathbf{z}_k) \cup S_k\}$. It is crucial to note that while performing a forward pass on a subset of tokens, the positional embeddings of these tokens in the actual sequence are preserved. Below we discuss the attention bias used in the forward pass.

Let $D_k = \mathcal{C}(\mathbf{z}_k)$ be the set of position indices of tokens decoded prior to step $k$. Importantly, we do not need to make any distinction between tokens decoded in the diffusion phase or those decoded in the sequential phase. This flexibility allows our sampler to use any denoising schedule $\mathcal{S}$.

Let $\sigma$ be the denoising ordering derived from $\mathcal{S}$. We define the $L \times L$ attention bias at step $k$ by

$$A_{i,j} = \begin{cases} 0 & \text{if } \sigma^{-1}(i) \geq \sigma^{-1}(j) \ \forall (i,j) \in (D_k \cup S_k) \times (D_k \cup S_k) \tag{17} \\ -\infty & \text{otherwise.} \tag{18} \end{cases}$$

Crucially, this is just standard causal attention applied to clean tokens generated prior to step $k$ and mask tokens to be decoded in step $k$, both sorted by $\sigma$; causal attention permits KV caching (Fig. 1).

## 5   EXPERIMENTS

We evaluate Eso-LMs on two standard language modeling benchmarks: the One Billion Words dataset (LM1B) (Chelba et al., 2014) and OpenWebText (OWT) (Gokaslan et al., 2019). We describe data processing, model architecture, training, and hardware details in Sec. C.3.

### 5.1   LIKELIHOOD EVALUATION

Our experiments show that Eso-LMs enable a **fine-grained interpolation between MDM and AR perplexities on LM1B and OWT** (Table 1 and Table 2) by adjusting $\alpha_0$ for training.

Table 1: Test perplexities (PPL; ↓) on LM1B for models trained for 1M steps. For diffusion models, we report PPL computed using the ELBO (8) as in prior work. *Reported in He et al. (2022). ¶No sentence packing. †Reported in Arriola et al. (2025). ‡Reported in Sahoo et al. (2025).

| | PPL (↓) | PPL (↓) (ELBO) |
|---|---|---|
| *Autoregressive (AR)* | | |
| Transformer‡ | 22.83 | |
| *Diffusion* | | |
| D3PM Uniform | | 137.90¶ |
| D3PM Absorb | | 76.90¶ |
| Diffusion-LM* | | 118.62¶ |
| DiffusionBert | | 63.78 |
| SEDD Absorb‡ | | 32.71¶ |
| SEDD Uniform¶ | | 40.25¶ |
| MDLM‡ | | 31.78 |
| UDLM‡ | | 36.71 |
| DUO‡ | | 33.68 |
| *Interpolating diffusion and AR* | | |
| BD3-LMs† | | |
| $L' = 16$ | | 30.60 |
| $L' = 8$ | | 29.83 |
| $L' = 4$ | | 28.23 |
| **Eso-LMs (Ours)** | | |
| $\alpha_0 = 1.0$ | | 35.00 |
| $\alpha_0 = 0.5$ | | 32.38 |
| $\alpha_0 = 0.25$ | | 29.14 |
| $\alpha_0 = 0.125$ | | 26.21 |
| $\alpha_0 = 0.0625$ | | 24.51 |

Table 2: Test perplexities (PPL; ↓) on OWT for models trained for 250K steps. For diffusion models, we report PPL computed using the ELBO (8) as with prior work. *For Eso-LMs, we also use importance-weighted bounds ($K = 100$) to get tight estimates of true PPLs (Sec. 3.3). †Denotes retrained models; for fair comparison, we did not finetune BD3-LMs from MDLM unlike in Arriola et al. (2025). ¶250K checkpoints were provided by Sahoo et al. (2024a); Schiff et al. (2025), or Sahoo et al. (2025).

| | PPL (↓) | PPL (↓) (ELBO) |
|---|---|---|
| *Autoregressive (AR)* | | |
| Transformer | 17.90¶ | |
| *Diffusion* | | |
| SEDD Absorb | | 26.81¶ |
| MDLM | | 25.19¶ |
| UDLM | | 30.52¶ |
| DUO | | 27.14¶ |
| *Interpolating diffusion and AR* | | |
| BD3-LMs | | |
| $L' = 16$ | | 23.57† |
| $L' = 8$ | | 22.04† |
| $L' = 4$ | | **20.96†** |
| **Eso-LMs (Ours)** | | |
| $\alpha_0 = 1$ | 29.80* | 30.06 |
| $\alpha_0 = 0.5$ | 27.09* | 27.85 |
| $\alpha_0 = 0.25$ | 23.56* | 24.73 |
| $\alpha_0 = 0.125$ | **20.86*** | **21.87** |

**Experimental Setup** We compare Eso-LMs against leading masked diffusion models—MDLM (Sahoo et al., 2024a), SEDD (Lou et al., 2024), D3PM (Austin et al., 2021), and DiffusionBERT (He et al., 2022)—as well as uniform state models DUO (Sahoo et al., 2025), UDLM (Schiff et al., 2025), and specifically BD3-LMs (Arriola et al., 2025), which also interpolate between MDM and AR and support KV caching. All models are trained with `batch_size=512`, consistent with prior work. We split each batch evenly: half trained with the AR loss and half with the MDM loss (8). Refer to Table 4 for an ablation on the split proportion $\kappa$. Refer to Algo. 2 for the training procedure. Attention biases are configured as described in Sec. 4. When training Eso-LMs as a pure MDM ($\alpha_0 = 1$), the full batch is trained with the MDM loss. For this setting only, we replace the diffusion coefficient $\alpha_t'/(1 - \alpha_t)$ with $-1$, which empirically reduced training variance and improved convergence.

**Results** For all diffusion models, PPL is computed using the lower bound (8) on the log-likelihood, following (Sahoo et al., 2024a; Schiff et al., 2025; Austin et al., 2021; Sahoo et al., 2024b; Lou et al., 2024; Arriola et al., 2025). We call this PPL (ELBO), an upper bound on PPL. On LM1B, we train Eso-LMs with $\alpha_0 \in \{0.0625, 0.125, 0.25, 0.5, 1.0\}$; we find that Eso-LMs effectively interpolate between MDLM and AR perplexities with $\alpha_0 \in \{0.0625, 0.125, 0.25, 0.5\}$ but exceeds MDLM PPL by ~ 3 points with $\alpha_0 = 1.0$ (Table 1). This is expected as Eso-LM ($\alpha_0 = 1$) is just MDLM but with sparse causal attention instead of bidirectional attention. Results hold similarly for OWT (Table 2).

**Importance-Weighted (IW) Bounds** To verify that the ordering of PPLs (ELBO) reflect the true ordering of PPLs for Eso-LMs, we use IW bounds ($K = 100$) (Sec. 3.3) to obtain tight estimates of PPLs, which we find to be close to the corresponding PPLs (ELBO) and fall in the same order (Table 2). This is the first time for IW bounds to be obtained for discrete diffusion. For diffusion baselines, IW bounds are intractable (Sec. 3.3). We include IW bounds for smaller $K$'s in Table 6.

**Ablation** Instead of fully switching from bidirectional to causal attention as in Eso-LMs (Sec. 4.1), we provide an intermediate ablation that mixes both. We name this family Eso-LMs (A) (details in Suppl. D). As shown in Table 7 and Table 8, Eso-LMs (A) also interpolate between MDLM and

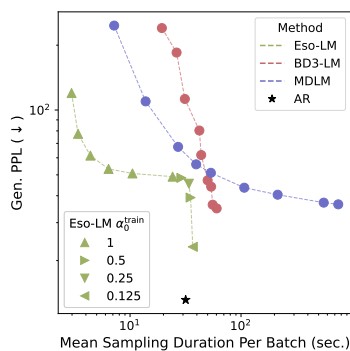 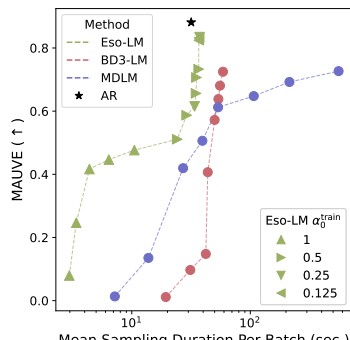

Figure 3: Eso-LMs establish SOTA on the Pareto frontier of sampling speed and Gen. PPL.

Figure 4: Eso-LMs establish SOTA on the Pareto frontier of sampling speed and MAUVE.

AR perplexities on LM1B and OWT. As expected, its perplexity is better than Eso-LMs at every $\alpha_0$, making its perplexity at $\alpha_0 = 1$ closer to MDLM, but it does not support KV caching during diffusion.

## 5.2 PARETO FRONTIER OF GENERATION SPEED VS. QUALITY

Our experiments show that (1) **Eso-LMs establish a new SOTA on the Pareto frontier of sampling speed and quality** (Fig. 3 and Fig. 4), and (2) **don't produce degenerate samples (poor quality and low diversity) at low NFEs** unlike the previous interpolating diffusion method BD3-LMs.

**Experimental Setup** We sample unconditionally from OWT models. We use Eso-LMs trained with $\alpha_0^{\text{train}} \in \{0.125, 0.25, 0.5, 1\}$ and generate samples by varying $(\alpha_0^{\text{eval}}, T) \in \{0.0625, 0.25, 0.5, 1\} \times \{16, 128, 1024\}$ to control NFEs (NFEs = $|\mathcal{S}|$) and sampling time. Although each model is trained with a single $\alpha_0^{\text{train}}$, we evaluate it across all inference-time $\alpha_0^{\text{eval}}$ values. MDLM and BD3-LMs use ancestral sampling as proposed in Sahoo et al. (2024a), with $T \in \{8, 16, 32, 64, 128, 256, 512, 1024, 4096\}$ for MDLM and $T \in \{128, 256, 512, 1024, 2048, 4096\}$ for BD3-LMs. All generations are $L = 1024$ tokens long. BD3-LMs are evaluated with block sizes $L' \in \{4, 8, 16\}$ and $T' = T/(1024/L')$; $T = 128$ is not applicable to BD3-LM with $L' = 4$ and $T = 16$ is not applicable to all BD3-LMs considered, since these would result in $T' < 1$. We measure Gen. Perplexity (via GPT-2 Large) and MAUVE (Pillutla et al., 2021) (via ModernBERT-Large) for sample quality and average entropy for diversity (Zheng et al., 2024), using nucleus sampling with $p = 0.9$ (Wang et al., 2025). Gen. PPL is a de facto metric used in prior work and MAUVE aligns with human judgments on open-ended text.

**Pareto Frontier of Generation Speed vs. Quality** We record the mean sampling duration in seconds (across 10 trials) by each method to generate a batch of 512 samples, and evaluate Gen. PPL and MAUVE using 5120 samples. Sampling duration is an increasing function of NFEs, modulated by the method and sampling hyperparameters used. In Fig. 3 and Fig. 4, for each method, we plot its speed-quality Pareto frontier over all its configurations: Eso-LMs (over $\alpha_0^{\text{train}}$, $\alpha_0^{\text{eval}}$, and $T$), BD3-LM (over $L'$ and $T$), and MDLM (over $T$). We find that Eso-LMs establish a new state of the art on the speed-quality Pareto frontier. See Sec. E.8 for individual metrics and Sec. E.9 for text samples.

**Best $\alpha_0$ for Training** We find that the Pareto frontier of the Eso-LM trained with $\alpha_0^{\text{train}} = 1$ is competitive with the Pareto frontier of all four trained Eso-LMs (Fig. 15 and Fig. 16). This shows that Eso-LMs trained for diffusion only can flexibly adapt to a diverse set of denoising schedules.

**Heuristic Improved Sampler** BD3-LMs suffer from a rapid drop in quality at low NFEs due to decoding close-by tokens in parallel (Sec. 6). Hence, given the flexibility of our sampler, we propose a heuristic sampler for Eso-LMs that strictly performs parallel decoding for tokens far apart (Sec. E.6). This sampler significantly improves Eso-LMs's generation quality at low NFEs (Fig. 17 and Fig. 18).

## 5.3 GENERATION LATENCY AT LONG CONTEXT

At longer contexts, **Eso-LMs are $3 - 4\times$ faster than prior diffusion based methods that support KV caching** and $14 - 65\times$ **faster than MDMs that don't support KV caching**.

**Experimental Setup** We compare inference times of our method, Eso-LMs, against MDLM and BD3-LMs with context lengths $L \in \{2048, 8192\}$, using the first-hitting sampler (Zheng et al., 2024),

and a batch size of 1. To simulate the worst-case scenario, we set $T \gg L$ to ensure all methods have approximately $L$ NFEs: $T = 1M$ for MDLM and Eso-LMs (for $T \gg L$, NFE is $L$ for all $\alpha_0^{\text{eval}}$'s), $T' = 5000$ (number of sampling steps per block) for BD3-LMs. We find that nucleus sampling yields a non-negligible overhead for all methods, and hence disable it to focus on speed vs. sequence length.

**Results** As shown in Table 9, as compared to MDLM which lacks KV caching, Eso-LMs is ~14× faster for $L = 2048$, and ~65× faster for $L = 8192$. Compared to BD3-LMs, which partially support caching, Eso-LMs are ~3.2× faster than BD3-LM ($L' = 16$) and ~3.8× faster than BD3-LM ($L' = 4$) at $L = 8192$. Additionally, we finetune Eso-LM ($\alpha_0^{\text{train}} = 0.125$) and BD3-LM ($L' = 4$), originally trained with $L = 1024$ (Sec. 5.1), for 1K steps with $L = 10240$ on OWT; as shown in Table 10, the Eso-LM produces similar quality samples while being 5× faster ($\alpha_0^{\text{eval}} = 0.125$, $T \gg L$).

These speedups stem from KV caching and the scheduler $\mathcal{S}$ that restricts the forward pass to the masked tokens that are supposed to be denoised and previously denoised clean tokens, avoiding redundant computation—a feature MDLM lacks completely and BD3-LMs lack for the current block under diffusion. As we restrict the NFEs to $L$, our method is slightly slower than AR models due to delayed KV reuse—only possible from the penultimate step (Fig. 1).

## 6 RELATED WORK, DISCUSSION, AND CONCLUSION

**AR models** AR models generate tokens left-to-right and remain state-of-the-art in quality, but suffer from slow, sequential inference and limited controllability. In contrast, Eso-LMs combine AR-like generation in a sequential phase with any-order, parallel generation in an initial diffusion phase. During diffusion, Eso-LMs support KV caching (Pope et al., 2022), previously exclusive to AR models, matching their inference speed. Its quality approaches AR models as the sequential phase increases.

**Masked diffusion** MDMs (Sahoo et al., 2024a; Shi et al., 2025) can generate multiple tokens per step but perform bidirectional attention over the entire context. Eso-LMs improve their efficiency in two ways. First, Eso-LMs restrict attention to clean and scheduled-to-denoise mask tokens only. Second, leveraging the connection to AO-ARMs (Ou et al., 2025), Eso-LMs replace bidirectional with causal attention to unlock KV caching. Though Eso-LMs may underperform MDLM in terms of perplexity (e.g., at $\alpha_0^{\text{train}} = 1$), they achieve a significantly better generation speed-quality tradeoff.

**Block diffusion** BD3-LMs (Arriola et al., 2025) use AR over blocks of tokens and apply MDM within each. They interpolate between AR and MDMs by changing block size, whereas Eso-LMs interpolate by varying the proportion of diffusion generation $\alpha_0$. Both support KV caching differently: BD3-LMs cache block-level conditioning, while Eso-LMs cache clean-token KV values across denoising steps. BD3-LMs' short blocks ($L' \leq 16$) significantly increase token conflicts (Liu et al., 2024); poor samples in one block also severely affect the sample quality of subsequent blocks due to the use of teacher forcing during training. Eso-LMs do not suffer from this problem.

**Concurrent work** Hu et al. (2025); Wu et al. (2025); Ma et al. (2025) also study KV caching for diffusion language models. There are two keys differences between our work and the aforementioned works. First, Eso-LMs perform a forward pass on a subset of token positions, while these methods perform a bidirectional forward pass over the entire context like MDLM. Second, Eso-LMs are trained end-to-end while concurrent methods rely on heuristics: they reuse KV values computed in previous steps as training-free approximations to KV values in the current step.

**Conclusion** We introduce a new paradigm for language modeling that fuses autoregressive (AR) models and masked diffusion models (MDMs), enabling seamless interpolation between the two in both generation speed and sample quality. Our method introduces KV caching in MDMs while preserving parallel generation, significantly accelerating inference. It outperforms block diffusion methods in both speed and accuracy, setting a new state of the art on language modeling benchmarks. Given we are working on language modeling, we carry the inherent risks and opportunities in this line of research.

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

CONTENTS

# Appendices

## APPENDIX A  BACKGROUND

### A.1  BD3-LMS HYPERPARAMETER $T'$ AND `NUM_TRIES`

In the original codebase of BD3-LMs (Arriola et al., 2025), the number of diffusion sampling steps $T'$ for each block is set to 5000. This is an extremely high $T'$ considering the fact that the number of tokens in each block $L'$ is at most 16. Having $L' \leq 16'$ and $T' = 5000$ means that off-the-shelf BD3-LMs are **not performing parallel generation** because tokens are almost always denoised one at a time.

Further, we found that BD3-LMs' codebase **cherry-picks its samples**. More specifically, to generate a single sample, the codebase keeps generating new samples (up to `num_tries` times) until one sample passes some quality-control test. By default, `num_tries = 10` and the codebase reports sampling failure when the 10 tries are exhausted with no samples passing the test. Empirically, we found that sampling failures don't occur for $T' = 5000$.

To investigate the true performance of BD3-LMs for parallel generation, we set `num_tries = 1`, disable the quality-control test and evaluate samples from BD3-LMs across a wide range of $T$ values (Fig. 5). Here and in Fig. 5, $T$ means the sum of sampling steps across all blocks for BD3-LMs, e.g., $L' = 16$ and $T = 4096$ means that $T' = 4096/(1024/16)) = 64$ sampling steps is used per block. In contrast, BD3-LMs' codebase uses $T' = 5000$ by default, which corresponds to $T = \infty$ in Figure Fig. 5. For MDLM, $T$ can be interpreted normally because it has no blocks.

As shown in Figure Fig. 5, as $T$ is decreased to enable more parallel generation, **both sample quality and sample diversity of BD3-LMs becomes significantly worse than MDLM** which is discussed in Sec. 6. We also found that increasing `num_tries` can somewhat improve the sample entropy of

BD3-LMs (second row of Table 3) and avoid degenerate samples, but doing so provides less or no improvements for AR and MDLM.

All five 1M-step checkpoints used in this section are publicly available Hugging Face checkpoints uploaded by BD3-LMs authors. In particular, their BD3-LM checkpoints are finetuned from MDLM.

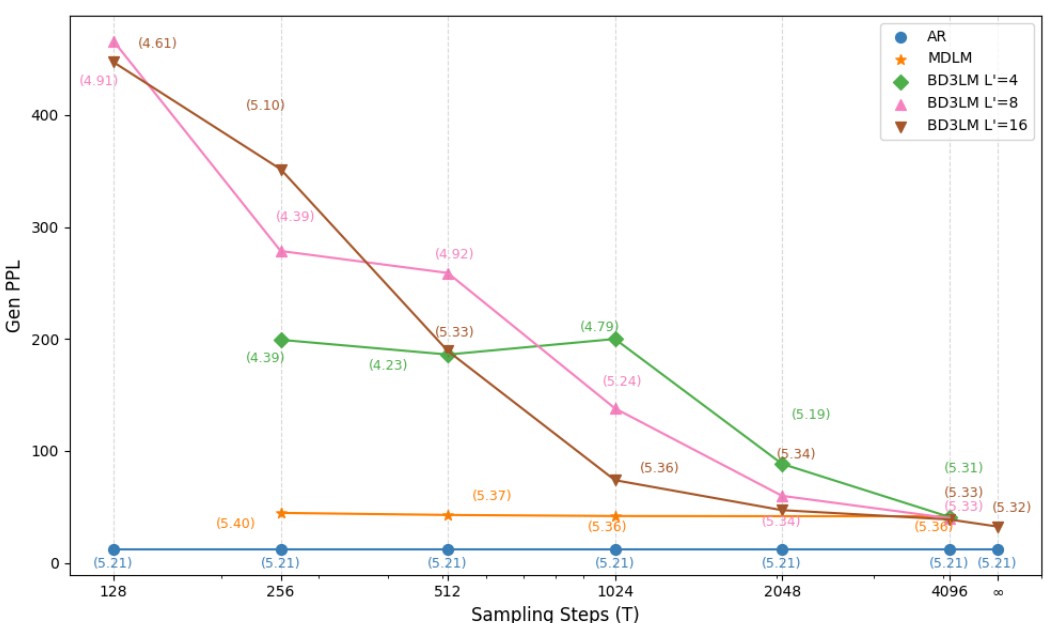

Figure 5: Gen. Perplexity ($\downarrow$) with nucleus sampling ($p = 0.9$) against the number of sampling steps for AR, MDLM and BD3-LMs trained for 1M steps. The number of sampling steps for AR is always 1024; we extend it to other values for easier comparison. The number next to each data point records its sample entropy ($\uparrow$); a value $< 5$ usually indicates low diversity degenerate samples.

Table 3: Gen. PPL ($\downarrow$) and entropy ($\uparrow$) (in parentheses) with nucleus sampling ($p = 0.9$) for AR, MDLM, and BD3-LM $L' = 16$ trained for 1M. We observe that the `num_tries` parameter introduced in (Arriola et al., 2025) for BD3-LMs selectively helps BD3-LMs but not the baselines. AR is not affected by $T$.

| | BD3-LM $L' = 16$ | | MDLM | | AR | |
|---|---|---|---|---|---|---|
| num_tries | 1 | 10 | 1 | 10 | 1 | 10 |
| $T = 1024$ | 72.80 (5.35) | 77.71 (5.41) | 41.92 (5.36) | 41.79 (5.37) | 13.03 (5.26) | 13.76 (5.32) |
| $T = 256$ | 356.02 (5.11) | 440.69 (5.28) | 45.07 (5.40) | 44.57 (5.39) | 13.03 (5.26) | 13.76 (5.32) |

## APPENDIX B    ESOTERIC LANGUAGE MODELS

### B.1    MDM LOSS DERIVATION

The NLL is given as:

$$-\log p_\theta(\mathbf{x}) \leq -\mathbb{E}_{\mathbf{z}_0 \sim q_0(.|\mathbf{x})} \log p_\theta^{\mathrm{AR}}(\mathbf{x}|\mathbf{z}_0) + \mathrm{D}_{\mathrm{KL}}(q_0(\mathbf{z}_0|\mathbf{x}) \| p_\theta^{\mathrm{MDM}}(\mathbf{z}_0))$$

$$= -\mathbb{E}_{\mathbf{z}_0 \sim q_0(.|\mathbf{x})} \left[ \sum_{\ell \in \mathcal{M}(\mathbf{z}_0)} \log p_\theta^{\mathrm{AR}}(\mathbf{x}^\ell|\mathbf{z}_0, \mathbf{x}^{<\ell}) \right] + \mathrm{D}_{\mathrm{KL}}(q_0(\mathbf{z}_0|\mathbf{x}) \| p_\theta^{\mathrm{MDM}}(\mathbf{z}_0)). \quad (19)$$

Note that $\mathbf{z}_0$ may contain clean tokens at indices exceeding the index $\ell$. As discussed in Sec. 3.1, the AR log-likelihood is given as:

$$\sum_{\ell \in \mathcal{M}(\mathbf{z}_0)} \log p_\theta^{\mathrm{AR}}(\mathbf{x}^\ell|\mathbf{z}_0, \mathbf{x}^{<\ell}) = \sum_{\ell \in \mathcal{M}(\mathbf{z}_0)} \log \langle \mathbf{x}_\theta^\ell(\mathbf{z}_0 \odot \mathbf{x}^{<\ell}), \mathbf{x}^\ell \rangle, \quad (20)$$

where we compute the loss only at the masked indices $\mathcal{M}(\mathbf{z}_0)$.

To compute the KL term in (19), we define a masked diffusion process over $\mathbf{z}_0$. For this diffusion process, its forward marginal $\mathbf{z}_t^\ell \sim q_t(\cdot|\mathbf{x}^\ell)$ at time $t \in [0,1]$ is the same as (1) but uses a noise schedule with a scaled-down range $(\alpha_t)_{t\in[0,1]} \in [0, \alpha_0]$, a strictly decreasing function in $t$ with $\alpha_{t=0} = \alpha_0$ such that $\mathbf{z}_{t=0} = \mathbf{z}_0$ and $\alpha_{t=1} = 0$ such that $\mathbf{z}_{t=1} = \mathbf{m}^{1:L}$. With $T$ diffusion steps, we have:

$$
\begin{aligned}
\mathrm{D_{KL}}(q_0(\mathbf{z}_0|\mathbf{x})\|p_\theta^{\mathrm{MDM}}(\mathbf{z}_0)) &= \mathbb{E}_{\mathbf{z}_0}\left[\log\frac{q_0(\mathbf{z}_0|\mathbf{x})}{p_\theta^{\mathrm{MDM}}(\mathbf{z}_0)}\right] \\
&= \mathbb{E}_{\mathbf{z}_0}\left[\log\mathbb{E}_{\mathbf{z}_{\frac{1}{T}:1}}\left[\frac{q(\mathbf{z}_{0:1}|\mathbf{x})}{p_\theta^{\mathrm{MDM}}(\mathbf{z}_{0:1})}\right]\right] \\
&\leq \mathbb{E}_{\mathbf{z}_{0:1}}\left[\log\frac{q(\mathbf{z}_{0:1}|\mathbf{x})}{p_\theta^{\mathrm{MDM}}(\mathbf{z}_{0:1})}\right] \\
&= \mathbb{E}_{\mathbf{z}_{0:1}}\left[\sum_{t\in\{\frac{1}{T},\frac{2}{T},\dots,1\}}\log\frac{q(\mathbf{z}_{t-\frac{1}{T}}|\mathbf{z}_t,\mathbf{x})}{p_\theta^{\mathrm{MDM}}(\mathbf{z}_{t-\frac{1}{T}}|\mathbf{z}_t)}\right] \\
&= \sum_t\mathbb{E}_{\mathbf{z}_t}\left[\mathrm{D_{KL}}(q(\mathbf{z}_{t-\frac{1}{T}}|\mathbf{z}_t,\mathbf{x})\|p_\theta^{\mathrm{MDM}}(\mathbf{z}_{t-\frac{1}{T}}|\mathbf{z}_t))\right]
\end{aligned}
$$

Sahoo et al. (2024a) show that, as $T \to \infty$, the above simplifies to:

$$
= \mathbb{E}_{t\sim\mathcal{U}[0,1],\mathbf{z}_t\sim q_t}\left[\frac{\alpha_t'}{1-\alpha_t}\sum_{\ell\in\mathcal{M}(\mathbf{z}_t)}\log\langle\mathbf{x}_\theta^\ell(\mathbf{z}_t),\mathbf{x}^\ell\rangle\right]. \tag{21}
$$

Finally, combining (20) and (21), we get the desired result:

$$
\mathcal{L}_{\mathrm{NELBO}}(\mathbf{x};\theta)
$$
$$
= \mathbb{E}_{\mathbf{z}_0\sim q_0}\left[\underbrace{-\sum_{\ell\in\mathcal{M}(\mathbf{z}_0)}\log\langle\mathbf{x}_\theta^\ell(\mathbf{z}_0\odot\mathbf{x}^{<\ell}),\mathbf{x}^\ell\rangle}_{\text{AR loss}}\right] + \underbrace{\int_{t=0}^{t=1}\frac{\alpha_t'}{1-\alpha_t}\mathbb{E}_{\mathbf{z}_t\sim q_t}\left[\sum_{\ell\in\mathcal{M}(\mathbf{z}_t)}\log\langle\mathbf{x}_\theta^\ell(\mathbf{z}_t),\mathbf{x}^\ell\rangle\right]\mathrm{d}t}_{\text{MDM loss}}.
$$
$$\tag{22}$$

## B.2 Training Algorithm

Algo. 2 outlines the complete training procedure.

---

**Algorithm 2** Eso-LMs Training

---

**Input:** dataset $D$, batch size $\mathtt{bs}$, forward noise process $q_t(\cdot|\mathbf{x})$, model $\mathbf{x}_\theta$, learning rate $\eta$
**while** not converged **do**
    $\mathbf{x}_1,\mathbf{x}_2,\dots,\mathbf{x}_{\mathtt{bs}} \sim D$
    **for** $i \leftarrow 1$ to $\mathtt{bs}/2$ **do**                                         $\triangleright$ If $\alpha_0 = 1$, loop through 1 to $\mathtt{bs}$.
        $\mathbf{z}_0 \sim q_0(\cdot|\mathbf{x})$
        $\sigma \sim \mathcal{P}_L$ with constraints        $\triangleright$ Used to construct the attention bias $A$ in $\mathbf{x}_\theta$ (Sec. 4)
        $\mathcal{L}_i \leftarrow -\sum_{\ell\in\mathcal{M}(\mathbf{z}_0)}\log\langle\mathbf{x}_\theta^\ell(\mathbf{z}_0,\mathbf{x}^{<\ell}),\mathbf{x}_i^\ell\rangle$    $\triangleright$ Estimator of Sequential Loss in (8)
    **end for**
    **for** $i \leftarrow \mathtt{bs}/2 + 1$ to $\mathtt{bs}$ **do**                                  $\triangleright$ If $\alpha_0 = 1$, skip this loop.
        Sample $t \sim \mathcal{U}[0,1]$
        $\mathbf{z}_t \sim q_t(\cdot|\mathbf{x})$
        $\sigma \sim \mathcal{P}_L$ with constraints        $\triangleright$ Used to construct the attention bias $A$ in $\mathbf{x}_\theta$ (Sec. 4)
        $\mathcal{L}_i \leftarrow \frac{\alpha_t'}{1-\alpha_t}\sum_{\ell\in\mathcal{M}(\mathbf{z}_t)}\log\langle\mathbf{x}_\theta^\ell(\mathbf{z}_t),\mathbf{x}_i^\ell\rangle$    $\triangleright$ Estimator of MDM Loss in (8)
    **end for**
    $\theta \leftarrow \theta - \eta\nabla_\theta\sum_{i=1}^{\mathtt{bs}}\mathcal{L}_i$
**end while**

---

## B.3 Denoising Schedule and Sampling Algorithm

Eso-LMs perform two phases of sampling: the diffusion phase and the sequential phase. Within the diffusion phase, tokens are denoised in random order and potentially in parallel. Within the sequential phase, remaining mask tokens are denoised sequentially from left to right and one at a time.

First, to determine (i) the total number of tokens to denoise during the diffusion phase and (ii) the number of tokens to denoise per diffusion step, we run a modified version of the first-hitting algorithm proposed in Zheng et al. (2024). Suppose the sequence to generate has length $L$, the number of discretization steps is $T$, and the noise schedule is $\alpha$ (with $\alpha_0 \geq 0$). Let $dt = 1/T$. We iterate from $t = 1$ to $1 - dt$ (inclusive) for $T$ steps. For each step, we compute the number of tokens to denoise at time $t$ as

$$n_t = \text{Binom}\left(n = n_t^{\text{remaining}}, p = \frac{\alpha_s - \alpha_t}{1 - \alpha_t}\right), \tag{23}$$

where $s = t - dt$ and $n_t^{\text{remaining}} = L - \sum_{t'>t} n_{t'}$. When $T$ is large, some $n_t$'s could be zero. All the $n_t$'s produced by this algorithm are collected in an ordered list, except for the $n_t$'s that are zeros. We denote the sum of all $n_t$'s as $n^{\text{MDM}}$ and define $n^{\text{AR}} = L - n^{\text{MDM}}$.

We select $n^{\text{MDM}}$ token indices from $[L]$ to denoise by diffusion and use the complementing subset of token indices to denoise sequentially. For example, suppose $L = 8$ and the token indices are $[1, 2, \ldots, 8]$. Suppose we obtained $n^{\text{MDM}} = 5$ from the algorithm above. Then, the diffusion indices we may select are $(1, 3, 4, 6, 7)$ and the complementing sequential indices are $(2, 5, 8)$. We further randomly permute the diffusion indices to be, e.g., $(3, 1, 6, 4, 7)$, for random-order denoising.

Given the list of non-zero $n_t$'s and the permuted ordered set of diffusion indices, we create the sampling schedule for diffusion by partitioning the diffusion indices per the $n_t$'s. Suppose the list of non-zero $n_t$'s is $(2, 1, 2)$. Using it to partition the permuted set of diffusion indices $(3, 1, 6, 4, 7)$, we obtain the following sampling schedule for the diffusion phase: $\mathcal{S}^{\text{MDM}} = ((3, 1), (6), (4, 7))$. The denoising schedule for the sequential phase is simply $\mathcal{S}^{\text{AR}} = ((2), (5), (8))$. The unified sampling schedule $\mathcal{S}$ is the concatenation of $\mathcal{S}^{\text{MDM}}$ and $\mathcal{S}^{\text{AR}}$. In this example, $\mathcal{S} = (S_1, S_2, S_3, S_4, S_5, S_6)$ where $S_1 = (3, 1), S_2 = (6), S_3 = (4, 7), S_4 = (2), S_5 = (5)$ and $S_6 = (8)$. This corresponds to 6 NFEs. Finally, $\mathcal{S}$ is passed to Algo. 3, which handles the rest of the sampling procedure. Connecting back to the denoising ordering $\sigma$ discussed in Sec. D.3 and Sec. 4.2, we have $\sigma = (3, 1, 6, 4, 7, 2, 5, 8)$ in this example.

---

**Algorithm 3** Eso-LMs Sampling

> **Input:** sequence length $L$, unified sampling schedule $\mathcal{S}$
> $\mathbf{z} = [\text{MASK\_INDEX}, \ldots, \text{MASK\_INDEX}]$
> $C = \{\}$                                     ▷ Indices of clean tokens
> **for** $i \leftarrow 1$ to $|\mathcal{S}|$ **do**          ▷ Sequential happens automatically when $|C| \geq n^{\text{MDM}}$
>     `logits` $\leftarrow \mathbf{x}_\theta(\mathbf{z}[C \cup S_i])$                      ▷ See *Remark*
>     `logits` $\leftarrow$ select `logits` corresponding to $S_i$
>     $\mathbf{z}[S_i] \leftarrow$ `categorical_sample(logits, dim=-1)`  ▷ `logits` has shape $(|S_i|, |\mathcal{V}|)$
>     $C \leftarrow C \cup S_i$
> **end for**
> **Return:** $\mathbf{z}$

*Remark.* $\mathbf{z}[C \cup S_i]$ denotes the subset of the tokens in $\mathbf{z}$ that are fed into the denoising model $\mathbf{x}_\theta$. The position embeddings for a token $\mathbf{z}^\ell \in \mathbf{z}[C \cup S_i]$ is ensured to be the same as that in the original sequence $\mathbf{z}$. Refer to Sec. D.3 and Sec. 4.2 for computing the sampling attention bias $A$ for Eso-LMs (A) and Eso-LMs respectively. For Eso-LMs, due to the use of causal attention, $\mathbf{x}_\theta$ is able to cache the KV-values of a clean token the first time it is processed.

---

## B.4 Attention Mechanism for Diffusion Phase Training

For a short and intuitive description, refer to Sec. 4.1.1.

In the diffusion phase, the denoising transformer receives $\mathbf{z}_t \sim q_t(.|\mathbf{x})$ as input, which contains mask tokens to denoise, and $\mathbf{x}$ as target. We leverage the connection of MDMs with AO-ARMs (Ou et al.,

2025), which establishes that mask tokens $\{\mathbf{z}_t^i | i \in \mathcal{M}(\mathbf{z}_t)\}$ can be denoised in any random order, and clean tokens $\{\mathbf{z}_t^i | i \in \mathcal{C}(\mathbf{z}_t)\}$ also could have been generated in any random order. Hence, we first sample a random ordering $\sigma \sim \mathcal{P}_L$ with the only constraint that clean tokens in $\mathbf{z}_t$ precede mask tokens in $\mathbf{z}_t$ per $\sigma$. We then constrain a clean token $(\mathbf{z}_t^i)_{i \in \mathcal{C}(\mathbf{z}_t)}$ to only attend to itself and prior clean tokens per $\sigma$; a mask token $(\mathbf{z}_t^i)_{i \in \mathcal{M}(\mathbf{z}_t)}$ attends to clean tokens, itself, and prior mask tokens per $\sigma$. Hence we define the $L \times L$ attention bias by

$$A_{i,j} = \begin{cases} 0 & \text{if } \sigma^{-1}(i) \geq \sigma^{-1}(j) \; \forall (i,j) \in [L] \times [L] \quad (24) \\ -\infty & \text{otherwise.} \quad (25) \end{cases}$$

See Fig. 6 for an example.

**Simplified Implementation**  $A$ becomes a causal attention bias if we sort the rows and columns of $A$ by $\sigma$ (Fig. 6), which is simple to implement. We also sort the positional embeddings of $\mathbf{z}_t$ by $\sigma$ so tokens keep their original positional embeddings. When calculating loss, we sort the target $\mathbf{x}$ by $\sigma$.

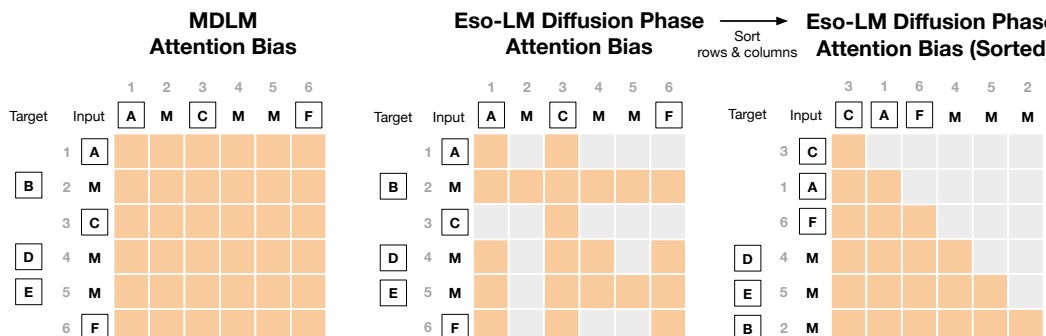

Figure 6: Comparison of attention biases for MDLM and Eso-LMs diffusion-phase training, before and after sorting the rows and columns by $\sigma$. **Orange** represents 0 (attention) and **gray** represents $-\infty$ (no attention). The clean sequence is $\mathbf{x} = (A, B, C, D, E, F)$ and hence $L = 6$. After random masking, we obtain $\mathbf{z}_t = (A, M, C, M, M, F)$. The integers denote position indices: $\mathcal{M}(\mathbf{z}_t) = \{2, 4, 5\}$ and $\mathcal{C}(\mathbf{z}_t) = \{1, 3, 6\}$. The ordering is $\sigma = (3, 1, 6, 4, 5, 2) \sim \mathcal{P}_6$ with clean tokens before mask tokens.

```python
from torch.nn.attention.flex_attention import create_block_mask

def _causal_mask(b, h, q_idx, kv_idx):
    causal = q_idx >= kv_idx
    return causal

def _get_causal_mask(seq_len):
    return create_block_mask(
        _causal_mask,
        B=None, H=None, Q_LEN=seq_len, KV_LEN=seq_len)
```

Figure 7: We implement the attention bias from Fig. 6 (Right) as a FlexAttention-compatible sparse masking function shown above that can handle arbitrary sequence lengths. This enables Just-In-Time compilation that's much more efficient than the naive scaled_dot_product_attention in PyTorch.

### B.5 ATTENTION MECHANISM FOR SEQUENTIAL PHASE TRAINING

For a precise mathematical definition of the attention mask, refer to Sec. 4.1.2.

See Fig. 8 for an illustrative example. See Fig. 9 for the PyTorch implementation.

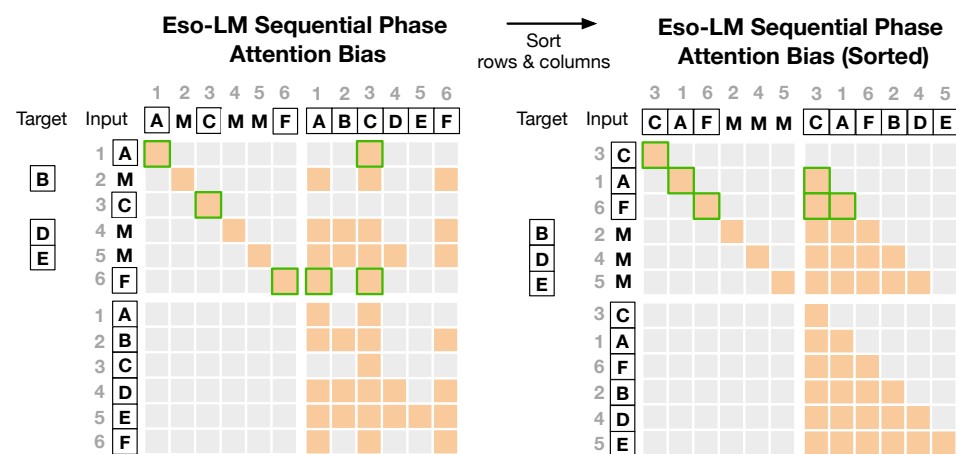

Figure 8: Comparison of attention biases for Eso-LMs sequential-phase training, before and after sorting the rows and columns of each of the four $L \times L$ blocks by $\sigma$. **Orange** represents 0 (attention) and **gray** represents $-\infty$ (no attention). The clean sequence is $\mathbf{x} = (A, B, C, D, E, F)$ and hence $L = 6$. After random masking, we obtain $\mathbf{z}_0 = (A, M, C, M, M, F)$. The integers denote the position indices with $\mathcal{M}(\mathbf{z}_0) = \{2, 4, 5\}$ and $\mathcal{C}(\mathbf{z}_0) = \{1, 3, 6\}$. The random ordering among $\mathcal{C}(\mathbf{z}_0)$ is $(3, 1, 6)$. **Green** highlights the extra connections added from clean tokens in $\mathbf{z}_0$ so that the attention bias display classic patterns after sorting — they don't contribute to the transformer output because no other token attends to clean tokens in $\mathbf{z}_0$.

```python
from torch.nn.attention.flex_attention import create_block_mask
from functools import partial

def _seq_mask(b, h, q_idx, kv_idx, n=None):
    # Indicate whether token belongs to zt or x
    x_flag_q = (q_idx >= n)
    x_flag_kv = (kv_idx >= n)

    # Adjust indices
    q_idx2 = torch.where(x_flag_q == 1, q_idx - n, q_idx)
    kv_idx2 = torch.where(x_flag_kv == 1, kv_idx - n, kv_idx)

    # 1. Diagonal Mask (Upper Left)
    diagonal = (q_idx2 == kv_idx2) & (x_flag_q == x_flag_kv)

    # 2. Offset Causal Mask (Upper Right)
    offset_causal = (q_idx2 > kv_idx2) & (x_flag_kv == 1) & (x_flag_q == 0)

    # 3. Causal Mask (Lower Right)
    causal = (q_idx2 >= kv_idx2) & (x_flag_kv == 1) & (x_flag_q == 1)

    # Combine the 3 masks together
    return diagonal | offset_causal | causal

def _get_seq_mask(seq_len):
    # Here, seq_len means the length of zt only
    return create_block_mask(
        partial(_seq_mask, n=seq_len),
        B=None, H=None, Q_LEN=seq_len*2, KV_LEN=seq_len*2)
```

Figure 9: We implement the attention bias from Fig. 8 (Right) as a FlexAttention-compatible sparse masking function shown above that can handle arbitrary sequence lengths. This enables Just-In-Time compilation that's much more efficient than the naive scaled_dot_product_attention in PyTorch.

### B.6 ATTENTION MECHANISM FOR SAMPLING

During sampling step $k$, given a partially masked sequence $\mathbf{z}_k$, the denoising model is required to denoise the mask tokens $\{\mathbf{z}_k^i | i \in S_k\}$ for $S_k \in \mathcal{S} = (S_1, \ldots, S_K)$ where $K = |\mathcal{S}|$. We perform a forward pass on the subset of tokens $\{\mathbf{z}_k^i | i \in \mathcal{C}(\mathbf{z}_k) \cup S_k\}$. It is crucial to note that while performing a forward pass on a subset of tokens, the positional embeddings of these tokens in the actual sequence are preserved. Below we discuss the attention bias used in the forward pass.

Let $D_k = \mathcal{C}(\mathbf{z}_k)$ be the set of position indices of tokens decoded prior to step $k$. Importantly, we do not need to make any distinction between tokens decoded in the diffusion phase or those decoded in the sequential phase. This flexibility allows our sampler to use any denoising schedule $\mathcal{S}$.

Let $\sigma$ be the denoising ordering derived from $\mathcal{S}$. We define the $L \times L$ attention bias at step $k$ by

$$A_{i,j} = \begin{cases} 0 & \text{if } \sigma^{-1}(i) \geq \sigma^{-1}(j) \ \forall (i,j) \in (D_k \cup S_k) \times (D_k \cup S_k) \quad (26) \\ -\infty & \text{otherwise,} \quad\quad\quad (27) \end{cases}$$

which is simply causal attention applied to clean tokens generated prior to step $k$ and mask tokens to be decoded in step $k$, both sorted by $\sigma$. Causal attention allows for KV caching, as shown in Fig. 10.

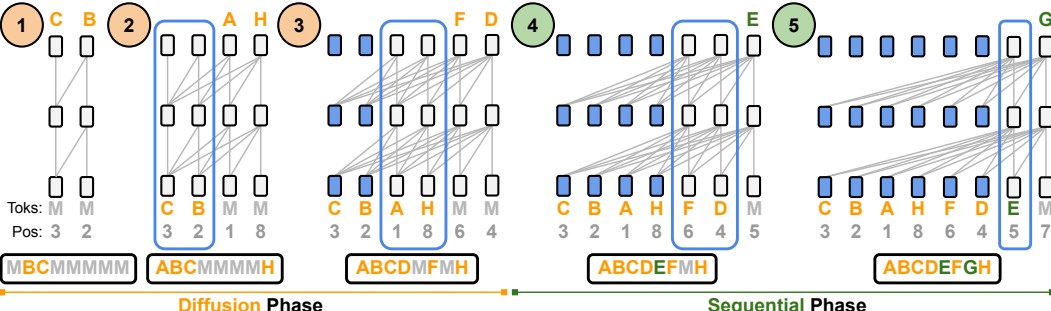

Figure 10: (Copy of Fig. 1) Efficient generation of an example sequence with Eso-LMs. During **Diffusion** Phase, Eso-LMs denoise one or more, potentially non-neighboring mask tokens ($\mathbb{M}$) per step. During **Sequential** Phase, Eso-LMs denoise the remaining mask tokens one at a time from left to right. Eso-LMs allows for **KV caching in both phases** using just **a single unified KV cache**: **blue** bounding boxes enclose transformer cells that are building their KV cache; a cell becomes **blue** once its KV cache is built. The sequences below the transformers depict tokens in their natural order.

## B.7 TRANSFORMER IMPLEMENTATION

```python
import torch.nn as nn

class Transformer(nn.Module):
  # ...
  def _get_attention_mask(self, diffusion_mode, seq_len):
    if diffusion_mode:
      return _get_causal_mask(seq_len)  # See Figure 7
    else:
      return _get_seq_mask(seq_len)  # See Figure 9

  def _sample_ordering(self, zt, shuffle_masks):
    masked = zt == self.mask_index
    offsets = torch.rand(zt.shape)
    if not shuffle_masks:
      # Induce left-to-right order within masked tokens
      offsets[masked] = torch.linspace(0, 1, torch.sum(masked))
    ordering = (masked + offsets).argsort(descending=False)
    return ordering

  def _sort(self, zt, ordering):
    return torch.gather(zt, dim=1, index=ordering)

  def forward(self, zt, x=None):
    '''
    x [batch size, L]: clean sequence (only for sequential training)
    zt [batch size, L]: randomly masked sequence
    '''

    seq_len = zt.shape[1]
    # Construct rotary embeddings for a given sequence
    rotary = self.rotary_emb(zt)  # [batch size, L, d]

    ### -- Start Extra Code --
    diffusion_mode = x is None
    attn_mask = self._get_attention_mask(diffusion_mode, seq_len)

    if diffusion_mode:  # Diffusion Mode
      # Shuffle as per [Line 309-312]
      # [batch size, L]
      ordering = self._sample_ordering(zt, shuffle_masks=True)
      x = self._sort(zt, ordering)
    else:  # Sequential Mode
      # Shuffle as per [Line 323-324]
      # [batch size, L]
      ordering = self._sample_ordering(zt, shuffle_masks=False)
      x = torch.cat([
        self._sort(x, ordering), self._sort(zt, ordering)], dim=1)
      rotary = self._sort(rotary, ordering)
      rotary = torch.cat([rotary, rotary], dim=1)
    ### -- End Extra Code --

    # Standard transformer forward pass
    for i in range(len(self.blocks)):
      x = self.transformer_blocks[i](
        x, rotary=rotary, attn_mask=attn_mask)
    logits = self.output_layer(x)

    # Logits will be compared against shuffled targets
    return logits, ordering
```

Figure 11: Eso-LMs introduce minimal changes to the Transformer architecture.

# APPENDIX C  EXPERIMENTAL DETAILS

## C.1  LOW DISCREPANCY SAMPLER

To reduce variance during training we use a low-discrepancy sampler, similar to that proposed Kingma et al. (2021). Specifically, when processing a minibatch of $N$ samples, instead of independently sampling $N$ from a uniform distribution, we partition the unit interval and sample the time step for each sequence $i \in \{1, \ldots, N\}$ from a different portion of the interval $t_i \sim U[\frac{i-1}{N}, \frac{i}{N}]$. This ensures that our sampled timesteps are more evenly spaced across the interval $[0, 1]$, reducing ELBO variance.

## C.2  LIKELIHOOD EVALUATION

We use a single monte-carlo estimate for $t$ for each example to evaluate the likelihood. We use a low discrepancy sampler (Kingma et al., 2021) to reduce the variance of the estimate.

## C.3  LANGUAGE MODELING

We detokenize the One Billion Words dataset following Lou et al. (2024); Sahoo et al. (2024a), whose code can be found here[1]. We tokenize the One Billion Words dataset with the `bert-base-uncased` tokenizer, following Austin et al. (2021); He et al. (2022). We concatenate and wrap sequences (also known as sequence packing) to a length of 128 (Raffel et al., 2020). When wrapping, we add the `[CLS]` token in-between concatenated sequences. The final preprocessed sequences also have the `[CLS]` token as their first and last token. Unlike Sahoo et al. (2024a); Lou et al. (2024); He et al. (2022), we apply sequence packing to LM1B, making our setup more challenging and resulting in higher perplexities given the same model (Table 1).

We tokenize OpenWebText with the `GPT2` (Radford et al., 2019) tokenizer. We concatenate and wrap them to a length of 1,024. When wrapping, we add the `eos` token in-between concatenated sequences. Unlike for One Billion Words, the final preprocessed sequences for OpenWebText do not have special tokens as their first and last token. Since OpenWebText does not have a test split, we leave the last 100k docs as test.

Eso-LMs shares the same parameterization as our autoregressive baseline, SEDD, MDLM, UDLM, and DUO: a modified diffusion transformer architecture (Peebles & Xie, 2023) from Lou et al. (2024); Sahoo et al. (2024a). We use 12 layers, a hidden dimension of 768, 12 attention heads. Eso-LMs do not use timestep embedding used in uniform diffusion models (SEDD Uniform, UDLM, DUO). Word embeddings are not tied between the input and output. We train BD3-LMs using the original code provided by their authors.

We use the log-linear noise schedule $\alpha_t = \alpha_0(1 - t)$. We use the AdamW optimizer with a batch size of 512, constant learning rate warmup from 0 to a learning rate of 3e-4 for 2,500 steps. We use a constant learning rate for 1M steps on One Billion Words and for 250K steps for OpenWebText. We use a dropout rate of 0.1. We train models on H200 GPUs. On OpenWebText for 250K steps, training takes ~27 hours when $\alpha_0 = 1$ and ~37 hours when $\alpha_0 < 1$ due to the additional AR loss. Throughput is benchmarked on H200 GPUs and latency is benchmarked on A6000 GPUs.

# APPENDIX D  ESO-LMS (A) AS AN ABLATION

## D.1  ATTENTION MECHANISM FOR DIFFUSION PHASE TRAINING

The denoising transformer receives $\mathbf{z}_t \sim q_t(.|\mathbf{x})$ as input, which contains the mask tokens to denoise, and $\mathbf{x}$ as target. A random ordering $\sigma \sim \mathcal{P}_L$ is sampled with the only constraint that clean tokens in $\mathbf{z}_t$ precede mask tokens in $\mathbf{z}_t$ in $\sigma$. We define the $L \times L$ attention bias by

$$A_{i,j} = \begin{cases} 0 & \forall (i,j) \in \mathcal{C}(\mathbf{z}_t) \times \mathcal{C}(\mathbf{z}_t) & (28) \\ 0 & \text{if } \sigma^{-1}(i) \geq \sigma^{-1}(j) \; \forall (i,j) \in \mathcal{M}(\mathbf{z}_t) \times [L] & (29) \\ -\infty & \text{otherwise.} & (30) \end{cases}$$

---

[1]https://github.com/louaaron/Score-Entropy-Discrete-Diffusion/blob/main/data.py

Clean tokens $\{\mathbf{z}_t^i | i \in \mathcal{C}(\mathbf{z}_t)\}$ have bidirectional attention among them (28), while a mask token $(\mathbf{z}_t^i)_{i \in \mathcal{M}(\mathbf{z}_t)}$ attends to clean tokens, itself and prior mask tokens per $\sigma$ (29). We can ignore the ordering among clean tokens in $\sigma$ due to the use of bidirectional attention. See Fig. 12 for an example.

**Simplified Implementation** $A$ becomes a Prefix-LM (Raffel et al., 2020) attention bias if we sort the rows and columns of $A$ by $\sigma$ (Fig. 6), which is simple to implement.

Figure 12: Comparing attention biases for MDLM and Eso-LMs (A) diffusion-phase training, before and after sorting the rows and columns by $\sigma$. **Orange** represents 0 (attention) and **gray** represents $-\infty$ (no attention). The clean sequence is $\mathbf{x} = (A, B, C, D, E, F)$ and hence $L = 6$. After random masking, we obtain $\mathbf{z}_t = (A, M, C, M, M, F)$. The integers denote position indices: $\mathcal{M}(\mathbf{z}_t) = \{2, 4, 5\}$ and $\mathcal{C}(\mathbf{z}_t) = \{1, 3, 6\}$. $\sigma = (3, 1, 6, 4, 5, 2) \sim \mathcal{P}_6$ with clean tokens before mask tokens.

## D.2 ATTENTION MECHANISM FOR SEQUENTIAL PHASE TRAINING

The denoising transformer receives the concatenated sequence $\mathbf{z}_0 \oplus \mathbf{x} \in \mathcal{V}^{2L}$ as input, where $\mathbf{z}_0 \sim q_0(.|\mathbf{x})$ contains the mask tokens to denoise, and $\mathbf{x}$ as target. We define the $2L \times 2L$ attention bias by

$$A_{i,j} = 0 \qquad \text{if } i = j \forall (i,j) \in \mathcal{M}(\mathbf{z}_0) \times \mathcal{M}(\mathbf{z}_0) \tag{31}$$

$$A_{i,j+L} = 0 \qquad \forall (i,j) \in \mathcal{M}(\mathbf{z}_0) \times \mathcal{C}(\mathbf{z}_0) \tag{32}$$

$$A_{i,j+L} = 0 \qquad \text{if } i > j \forall (i,j) \in \mathcal{M}(\mathbf{z}_0) \times \mathcal{M}(\mathbf{z}_0) \tag{33}$$

$$A_{i+L,j+L} = 0 \qquad \forall (i,j) \in \mathcal{C}(\mathbf{z}_0) \times \mathcal{C}(\mathbf{z}_0) \tag{34}$$

$$A_{i+L,j+L} = 0 \qquad \forall (i,j) \in \mathcal{M}(\mathbf{z}_0) \times \mathcal{C}(\mathbf{z}_0) \tag{35}$$

$$A_{i+L,j+L} = 0 \qquad \text{if } i \geq j \forall (i,j) \in \mathcal{M}(\mathbf{z}_0) \times \mathcal{M}(\mathbf{z}_0) \tag{36}$$

$$A_{i,j} = -\infty \qquad \text{otherwise.} \tag{37}$$

See Fig. 13 for an example. This construction ensures that a mask token $(\mathbf{z}_0^i)_{i \in \mathcal{M}(\mathbf{z}_0)}$ attends to (i) itself (31), (ii) the clean tokens $\{\mathbf{x}^j | j \in \mathcal{C}(\mathbf{z}_0)\}$ (32) and (iii) the clean versions of mask tokens on its left $\{\mathbf{x}^j | j \in \mathcal{M}(\mathbf{z}_0), i > j\}$ (33). A clean token $(\mathbf{z}_0^i)_{i \in \mathcal{C}(\mathbf{z}_0)}$ can attend to anything because no other token attends to them (37). The attention mechanism for tokens in the clean context $\mathbf{x}_0$ is described as follows. Tokens $\{\mathbf{x}^i | i \in \mathcal{C}(\mathbf{z}_0)\}$ have bidirectional attention (34). A clean token corresponding to a mask token, $(\mathbf{x}^i)_{i \in \mathcal{M}(\mathbf{z}_0)}$, attends to $\{\mathbf{x}^j | j \in \mathcal{C}(\mathbf{z}_0)\}$ (35) and $\{\mathbf{x}^j | j \in \mathcal{M}(\mathbf{z}_0), i \geq j\}$ (36).

**Simplified Implementation** Let $\sigma$ be an ordering such that: (i) clean tokens in $\mathbf{z}_0$ precede mask tokens in $\mathbf{z}_0$ in $\sigma$ and (ii) mask tokens in $\mathbf{z}_0$ are in natural order in $\sigma$. The ordering among clean tokens $\{\mathbf{x}^i | i \in \mathcal{C}(\mathbf{z}_0)\}$ can be ignored with bidirectional attention. When the rows and columns of each of the four $L$-by-$L$ blocks are sorted by $\sigma$, $A$ shows classic attention patterns (Fig. 13) that are simple to implement.

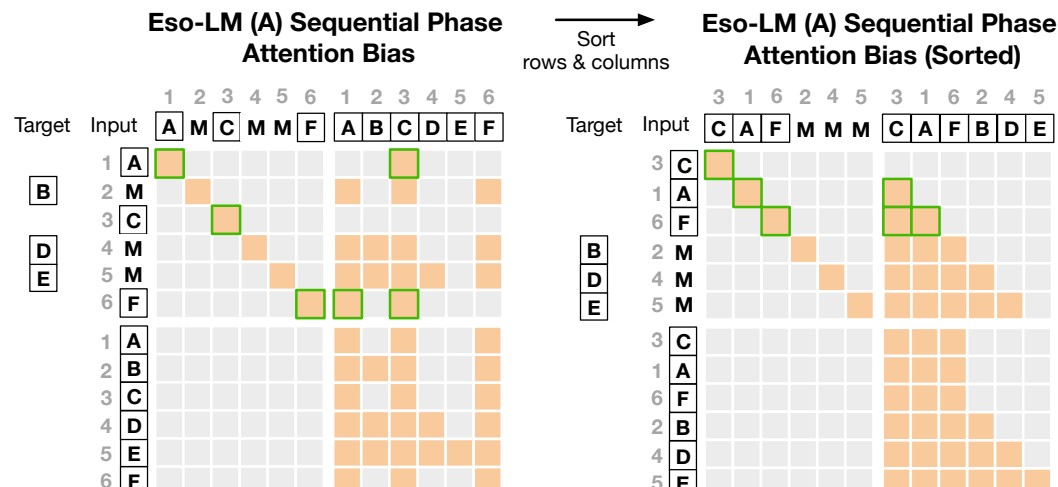

Figure 13: Comparison of attention biases for Eso-LMs (A) sequential-phase training, before and after sorting the rows and columns of each of the four $L \times L$ blocks by $\sigma$. **Orange** represents 0 (attention) and **gray** represents $-\infty$ (no attention). The clean sequence is $\mathbf{x} = (A, B, C, D, E, F)$ and hence $L = 6$. After random masking, we obtain $\mathbf{z}_0 = (A, M, C, M, M, F)$. The integers denote the position indices with $\mathcal{M}(\mathbf{z}_0) = \{2, 4, 5\}$ and $\mathcal{C}(\mathbf{z}_0) = \{1, 3, 6\}$. The random ordering among $\mathcal{C}(\mathbf{z}_0)$ is $(3, 1, 6)$. **Green** highlights the extra connections added from clean tokens in $\mathbf{z}_0$ so that the attention bias display classic patterns after sorting — they don't contribute to the transformer output because no other token attends to clean tokens in $\mathbf{z}_0$.

### D.3    ATTENTION MECHANISM FOR SAMPLING

During diffusion or sequential sampling, given a partially masked sequence $\mathbf{z}_k$, the denoising model is required to denoise the mask tokens $\{\mathbf{z}_k^i | i \in S_k\}$ for $S_k \in \mathcal{S} = (S_1, \dots, S_K)$ where $K = |\mathcal{S}|$. We perform a forward pass on the subset of tokens $\{\mathbf{z}_k^i | i \in \mathcal{C}(\mathbf{z}_k) \cup S_k\}$. It is crucial to note that while performing a forward pass on a subset of tokens, the positional embeddings of these tokens in the actual sequence are preserved. Below we discuss the attention bias used in the forward pass.

Let $D_k^{\mathrm{MDM}}$ be the set of indices of tokens decoded in the diffusion phase prior to step $k$ and $D_k^{\mathrm{AR}}$ be that for the sequential phase. Let ordering $\sigma$ be the order in which we denoise tokens defined by $\mathcal{S}$. We define the $L \times L$ attention bias at step $k$ by

$$
A_{i,j} = \begin{cases}
0 & \forall (i,j) \in D_k^{\mathrm{MDM}} \times D_k^{\mathrm{MDM}} & (38) \\
0 & \forall (i,j) \in D_k^{\mathrm{AR}} \times D_k^{\mathrm{MDM}} & (39) \\
0 & \text{if } i \geq j \ \forall (i,j) \in D_k^{\mathrm{AR}} \times D_k^{\mathrm{AR}} & (40) \\
0 & \forall (i,j) \in S_k \times (D_k^{\mathrm{MDM}} \cup D_k^{\mathrm{AR}}) & (41) \\
0 & \text{if } \sigma^{-1}(i) \geq \sigma^{-1}(j) \ \forall (i,j) \in S_k \times S_k & (42) \\
-\infty & \text{otherwise.} & (43)
\end{cases}
$$

Clean tokens decoded during diffusion $\{\mathbf{z}_k^i | i \in D_k^{\mathrm{MDM}}\}$ have bidirectional attention among them (38). A clean token decoded sequentially $(\mathbf{z}_k^i)_{i \in D_k^{\mathrm{AR}}}$ attends to clean tokens decoded during diffusion $\{\mathbf{z}_k^j | j \in D_k^{\mathrm{MDM}}\}$ (39), itself, and prior clean tokens decoded sequentially $\{\mathbf{z}_k^j | j \in D_k^{\mathrm{AR}}, i > j\}$ (40). A mask token to denoise $(\mathbf{z}_k^i)_{i \in S_k}$ attends to all decoded clean tokens $\{\mathbf{z}_k^j | j \in D_k^{\mathrm{MDM}} \cup D_k^{\mathrm{AR}}\}$ (41), itself, and prior mask tokens to denoise per $\sigma$: $\{\mathbf{z}_k^j | j \in S_k, \sigma^{-1}(i) > \sigma^{-1}(j)\}$ (42). Mask tokens not scheduled to denoise $(\mathbf{z}_k^i)_{i \in S_{>k}}$ can attend to anything because no other token attends to them (43).

Fig. 14 shows how Eso-LMs (A) generates with KV caching only during the sequential phase.

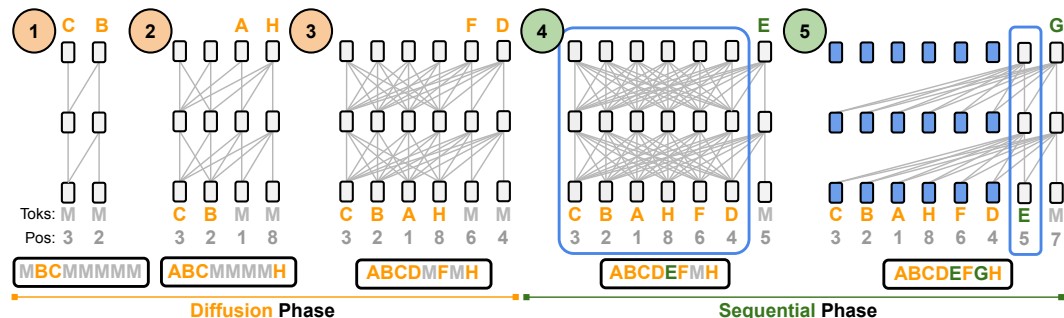

Figure 14: Generation of an example sequence with Eso-LMs (A). During **Diffusion** Phase, Eso-LMs denoise one or more, potentially non-neighboring mask tokens (M) per step. During **Sequential** Phase, Eso-LMs denoise the remaining mask tokens one at a time from left to right. Eso-LMs (A) allows for **KV caching in sequential phase** only: **blue** bounding boxes enclose transformer cells that are building their KV cache; a cell becomes **blue** once its KV cache is built. The sequences below the transformers depict tokens in their natural order.

# APPENDIX E  ADDITIONAL EXPERIMENTS

## E.1  ABLATION ON SPLIT PROPORTION

See Table 4.

Table 4: Test perplexities (↓) on LM1B for Eso-LMs (A) trained for 500K vs. the proportion $\kappa$ of examples in each batch used for evaluating the MDM loss in (8) during training. Remaining examples in each batch are used for evaluating the AR loss in (8) during training.

|  | $\kappa = 0.75$ | $\kappa = 0.5$ | $\kappa = 0.25$ | $\kappa = 0.125$ |
|---|---|---|---|---|
| Eso-LMs (A) |  |  |  |  |
| $\alpha_0 = 0.5$ | 32.25 | **31.53** | Diverged | Diverged |
| $\alpha_0 = 0.25$ | 30.49 | **29.33** | Diverged | Diverged |
| $\alpha_0 = 0.125$ | 27.76 | **26.73** | Diverged | Diverged |
| $\alpha_0 = 0.0625$ | 25.92 | **25.07** | Diverged | Diverged |

## E.2  ZERO-SHOT LIKELIHOOD EVALUATION

We explore models' ability to generalize by taking models trained on OWT and evaluating how well they model unseen datasets (Table 5). We compare the perplexities of our Eso-LMs with SEDD (Austin et al., 2021), MDLM (Sahoo et al., 2024a), BD3-LMs (Arriola et al., 2025), and an AR Transformer language model. Our zero-shot datasets are validation splits of Penn Tree Bank (PTB; (Marcus et al., 1993)), Wikitext (Merity et al., 2016), LM1B, Lambada (Paperno et al., 2016), AG News (Zhang et al., 2015), and Scientific Papers (Pubmed and Arxiv subsets; (Cohan et al., 2018)).

Table 5: Zero-shot perplexities (↓) of models trained for 250K steps on OWT. We report bounds for diffusion models and interpolation methods. Numbers for AR were taken from (Arriola et al., 2025).

|  | PTB | Wikitext | LM1B | Lambada | AG News | Pubmed | Arxiv |
|---|---|---|---|---|---|---|---|
| AR | 81.07 | 25.32 | 51.14 | 52.13 | 52.11 | 48.59 | 41.22 |
| MDLM | 93.82 | 36.89 | 69.45 | 53.05 | 67.33 | 49.47 | 43.84 |
| SEDD Absorb | 99.59 | 38.55 | 72.51 | 52.16 | 72.62 | 47.07 | 41.18 |
| BD3-LM ($L' = 16$) | 90.63 | 33.14 | 64.88 | 53.09 | 62.5 | 43.25 | 39.82 |
| **Eso-LMs (Ours)** | | | | | | | |
| $\alpha_0 = 1$ | 126.29 | 45.08 | 82.01 | 61.37 | 98.22 | 62.37 | 55.76 |
| $\alpha_0 = 0.5$ | 110.70 | 39.57 | 75.75 | 57.33 | 86.65 | 60.20 | 53.78 |
| $\alpha_0 = 0.25$ | 105.19 | 37.32 | 67.69 | 60.15 | 75.74 | 62.45 | 55.31 |
| $\alpha_0 = 0.125$ | 97.46 | 35.65 | 60.11 | 69.13 | 65.26 | 65.27 | 57.4 |

## E.3 IMPORTANCE-WEIGHTED BOUNDS

See Table 6.

Table 6: Test perplexities (↓) on OWT for Eso-LMs trained for 250K steps, computed using importance-weighted bounds. We report multiple estimates for each $\alpha_0$ by varying the number of orderings sampled ($K \in \{1, 10, 20, 50, 100\}$) per batch of 32 examples in the OWT test set.

|  | $K = 1$ | $K = 10$ | $K = 20$ | $K = 50$ | $K = 100$ |
|---|---|---|---|---|---|
| **Eso-LMs (Ours)** | | | | | |
| $\alpha_0 = 1$ | 31.71 | 30.50 | 30.26 | 29.99 | **29.80** |
| $\alpha_0 = 0.5$ | 28.95 | 27.77 | 27.53 | 27.27 | **27.09** |
| $\alpha_0 = 0.25$ | 25.23 | 24.16 | 23.95 | 23.72 | **23.56** |
| $\alpha_0 = 0.125$ | 22.24 | 21.35 | 21.17 | 20.98 | **20.86** |

## E.4 ESO-LMS (A) LIKELIHOOD EVALUATION

See Table 7 and Table 8.

Table 7: Test perplexities (↓) on LM1B for Eso-LMs and Eso-LMs (A) trained for 1M steps.

| $\alpha_0$ | Eso-LMs | Eso-LMs (A) |
|---|---|---|
| 1.0 | 35.00 | 30.96 |
| 0.5 | 32.38 | 30.51 |
| 0.25 | 29.14 | 28.44 |
| 0.125 | 26.21 | 25.97 |
| 0.0625 | **24.51** | **24.51** |

Table 8: Test perplexities (↓) on OWT for Eso-LMs and Eso-LMs (A) trained for 250K steps.

| $\alpha_0$ | Eso-LMs | Eso-LMs (A) |
|---|---|---|
| 1.0 | 30.06 | 26.21 |
| 0.5 | 27.85 | 25.38 |
| 0.25 | 24.73 | 23.78 |
| 0.125 | **21.87** | **21.47** |

## E.5 PARETO FRONTIER OF ESO-LMS WITH $\alpha_0^{\text{TRAIN}} = 1$

See Fig. 15 and Fig. 16 for a comparison of the Pareto frontier of Eso-LMs trained with $\alpha_0^{\text{train}} = 1$ against Pareto frontiers reported in the main paper (Fig. 3 and Fig. 4).

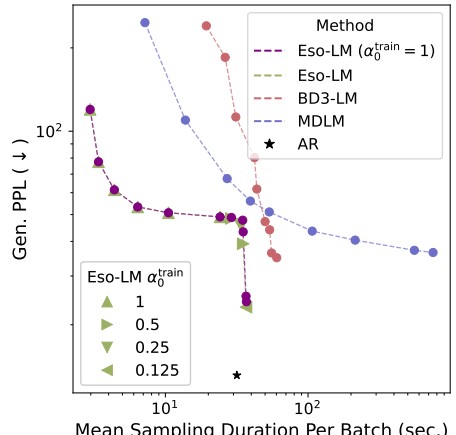

Figure 15: Eso-LMs establish SOTA on the Pareto frontier of sampling speed and Gen. PPL.

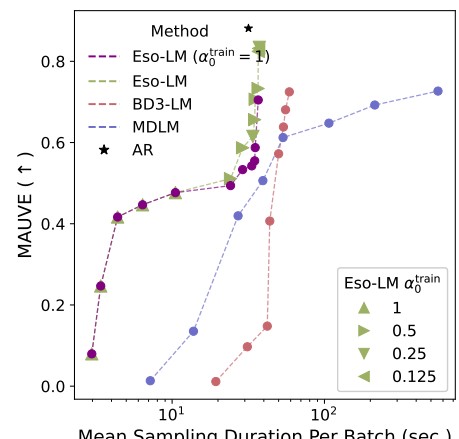

Figure 16: Eso-LMs establish SOTA on the Pareto frontier of sampling speed and MAUVE.

### E.6 HEURISTIC IMPROVED SAMPLER

We propose a heuristic improved sampler that only performs parallel decoding for evenly spaced positions across the sequence length. For example, with length 1024 and parallelism 4, the model first predicts positions 0, 255, 511, and 767 simultaneously. Subsequent steps need not target adjacent indices (e.g., 1, 256, 512, and 768), but instead continue to perform parallel decoding for a random set of 4 interleaved, far-apart positions. This process is iterated until the sequence is filled.

We use Eso-LMs trained with $\alpha_0^{\text{train}} = 1$ and generate samples by fixing $\alpha_0^{\text{eval}} = 1$ and varying $T$ to control NFEs and sampling time. For the improved sampler, we use Eso-LMs trained with $\alpha_0^{\text{train}} = 1$ and generate samples by varying the amount of parallelism, i.e., number of tokens generated in parallel: $\{64, 32, 16, 8, 4, 2, 1\}$. We find that the sampler significantly improves generation quality at low NFEs (Fig. 17 and Fig. 18) while offering less improvements at high NFEs, which is expected.

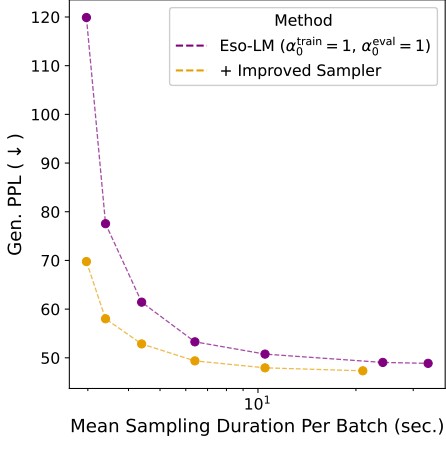

Figure 17: Heuristic improved sampler improves Gen. PPL Pareto frontier at low NFEs.

Figure 18: Heuristic improved sampler improves MAUVE Pareto frontier at low NFEs.

### E.7 GENERATION LATENCY AT LONG CONTEXT

Table 9: Sampling time ($\downarrow$) in seconds for sequence lengths $L \in \{2048, 8192\}$ with NFEs set to $L$ for all methods. Reported values are $\text{mean}_{\text{std}}$ over 5 runs.

| Method | $L = 2048$ | $L = 8192$ |
|---|---|---|
| **AR** | $13.3_{0.9}$ | $54.0_{0.2}$ |
| MDLM | $201.3_{0.4}$ | $5438.3_{3.3}$ |
| BD3-LMs ($L' = 4$) | $24.3_{0.7}$ | $312.0_{1.7}$ |
| BD3-LMs ($L' = 16$) | $21.3_{0.1}$ | $268.1_{1.2}$ |
| **Eso-LMs (Ours)** | $\mathbf{14.6_{0.3}}$ | $\mathbf{82.1_{0.3}}$ |

Table 10: Gen. PPL ($\downarrow$), entropy, and sampling time ($\downarrow$) in seconds for sequence length $L = 10240$ with NFEs set to $L$ for all methods. Reported values for sampling time are $\text{mean}_{\text{std}}$ over 5 runs.

| Method | Gen. PPL | Entropy | Time (seconds) |
|---|---|---|---|
| BD3-LMs ($L' = 4$) | 29.50 | 6.5 | $\mathbf{588.6_{3.2}}$ |
| **Eso-LM (Ours)** ($\alpha_0^{\text{train}} = \alpha_0^{\text{eval}} = 0.125$) | **23.40** | 6.3 | $\mathbf{116.4_{0.4}}$ |

### E.8 QUALITY OF GENERATED SAMPLES BY MODELS TRAINED ON OWT

In Fig. 3 and Fig. 4 we present how the sample quality changes by varying NFEs. The individual values for Gen. PPL, entropy and MAUVE can be found in Table 11 (Eso-LMs), Table 12 (MDLM), and Table 13 (BD3-LMs).

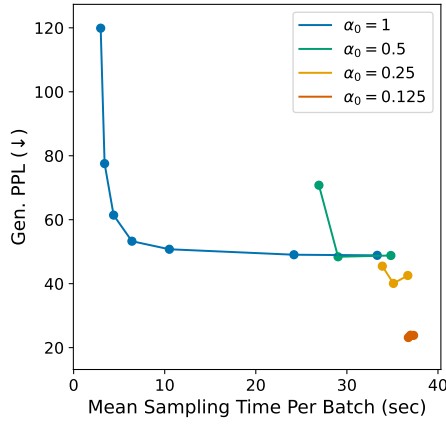

Figure 19: Decomposing the Pareto frontier on sampling speed and Gen. PPL of Eso-LMs into individual frontiers where $\alpha_0^{\text{train}} = \alpha_0^{\text{eval}}$ (or $\approx$).

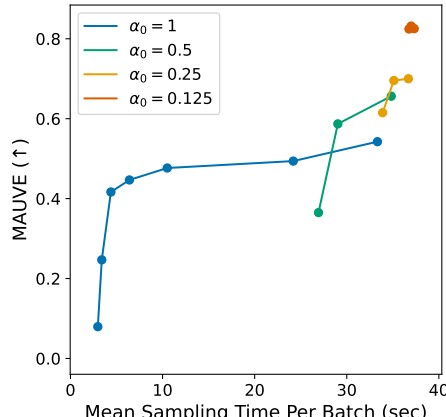

Figure 20: Decomposing the Pareto frontier on sampling speed and MAUVE of Eso-LMs into individual frontiers where $\alpha_0^{\text{train}} = \alpha_0^{\text{eval}}$ (or $\approx$).

Table 11: Gen. PPL (↓), entropies (↑), and MAUVE (↑) of samples by Eso-LMs trained for 250K steps on OWT.

| $\alpha_0^{\text{train}}$ | $\alpha_0^{\text{eval}}$ | $T$ | NFE | Gen. PPL (↓) | Entropy | MAUVE (↑) | Sampling Time (sec) (↓) |
|---|---|---|---|---|---|---|---|
| 1 | 0.0625 | 16 | 976 | 25.36 | 5.1 | 0.7048 | 36.75 |
| 1 | 0.0625 | 128 | 1010 | 24.74 | 5.1 | 0.6753 | 37.32 |
| 1 | 0.0625 | 1024 | 1022 | 24.23 | 5.1 | 0.6925 | 36.99 |
| 1 | 0.25 | 16 | 784 | 51.11 | 5.4 | 0.4996 | 33.89 |
| 1 | 0.25 | 128 | 879 | 43.31 | 5.3 | 0.5875 | 35.11 |
| 1 | 0.25 | 1024 | 994 | 43.36 | 5.3 | 0.5748 | 36.69 |
| 1 | 0.5 | 16 | 529 | 72.16 | 5.5 | 0.2885 | 26.93 |
| 1 | 0.5 | 128 | 639 | 48.80 | 5.3 | 0.5333 | 29.03 |
| 1 | 0.5 | 1024 | 913 | 47.72 | 5.3 | 0.5549 | 34.83 |
| 1 | 1 | 16 | 16 | 119.89 | 5.5 | 0.0796 | 2.97 |
| 1 | 1 | 32 | 32 | 77.55 | 5.5 | 0.2468 | 3.40 |
| 1 | 1 | 64 | 64 | 61.43 | 5.4 | 0.4166 | 4.39 |
| 1 | 1 | 128 | 128 | 53.28 | 5.4 | 0.4467 | 6.40 |
| 1 | 1 | 256 | 251 | 50.76 | 5.3 | 0.4766 | 10.51 |
| 1 | 1 | 1024 | 646 | 49.05 | 5.3 | 0.4939 | 24.19 |
| 1 | 1 | 4096 | 906 | 48.86 | 5.3 | 0.5425 | 33.33 |
| 0.5 | 0.0625 | 16 | 976 | 27.52 | 5.3 | 0.7905 | 36.75 |
| 0.5 | 0.0625 | 128 | 1010 | 27.84 | 5.3 | 0.8227 | 37.32 |
| 0.5 | 18 | 1024 | 1022 | 27.90 | 5.3 | 0.8160 | 36.99 |
| 0.5 | 0.25 | 16 | 784 | 45.81 | 5.4 | 0.5998 | 33.89 |
| 0.5 | 0.25 | 128 | 879 | 39.22 | 5.4 | 0.7066 | 35.11 |
| 0.5 | 0.25 | 1024 | 994 | 40.50 | 5.4 | 0.7330 | 36.69 |
| 0.5 | 0.5 | 16 | 529 | 70.78 | 5.5 | 0.3651 | 26.93 |
| 0.5 | 0.5 | 128 | 639 | 48.41 | 5.4 | 0.5870 | 29.03 |
| 0.5 | 0.5 | 1024 | 913 | 48.81 | 5.4 | 0.6563 | 34.83 |
| 0.5 | 1 | 16 | 16 | 125.21 | 5.5 | 0.0701 | 2.97 |
| 0.5 | 1 | 32 | 32 | 81.37 | 5.5 | 0.2118 | 3.40 |
| 0.5 | 1 | 64 | 64 | 64.04 | 5.4 | 0.3534 | 4.39 |
| 0.5 | 1 | 128 | 128 | 56.64 | 5.4 | 0.4232 | 6.40 |
| 0.5 | 1 | 256 | 251 | 53.53 | 5.4 | 0.4564 | 10.51 |
| 0.5 | 1 | 1024 | 646 | 53.24 | 5.4 | 0.5110 | 24.19 |
| 0.5 | 1 | 4096 | 906 | 54.11 | 5.4 | 0.5315 | 33.33 |
| 0.25 | 0.0625 | 16 | 976 | 24.20 | 5.4 | 0.7908 | 36.75 |
| 0.25 | 0.0625 | 128 | 1010 | 25.48 | 5.4 | 0.8344 | 37.32 |
| 0.25 | 0.0625 | 1024 | 1022 | 25.97 | 5.4 | 0.8312 | 36.99 |
| 0.25 | 0.25 | 16 | 784 | 45.48 | 5.4 | 0.6151 | 33.89 |
| 0.25 | 0.25 | 128 | 879 | 40.08 | 5.4 | 0.6955 | 35.11 |
| 0.25 | 0.25 | 1024 | 994 | 42.56 | 5.4 | 0.7000 | 36.69 |
| 0.25 | 0.5 | 16 | 529 | 79.84 | 5.5 | 0.1846 | 26.93 |
| 0.25 | 0.5 | 128 | 639 | 56.05 | 5.4 | 0.4125 | 29.03 |
| 0.25 | 0.5 | 1024 | 913 | 58.20 | 5.4 | 0.4558 | 34.83 |
| 0.25 | 1 | 16 | 16 | 154.93 | 5.5 | 0.0289 | 2.97 |
| 0.25 | 1 | 32 | 32 | 103.39 | 5.5 | 0.0798 | 3.40 |
| 0.25 | 1 | 64 | 64 | 82.31 | 5.4 | 0.1412 | 4.39 |
| 0.25 | 1 | 128 | 128 | 73.17 | 5.4 | 0.1801 | 6.40 |
| 0.25 | 1 | 256 | 251 | 69.82 | 5.4 | 0.1967 | 10.51 |
| 0.25 | 1 | 1024 | 646 | 71.42 | 5.4 | 0.2491 | 24.19 |
| 0.25 | 1 | 4096 | 906 | 74.39 | 5.4 | 0.2410 | 33.33 |
| 0.125 | 0.0625 | 16 | 976 | 23.16 | 5.4 | 0.8245 | 36.75 |
| 0.125 | 0.0625 | 128 | 1010 | 23.83 | 5.4 | 0.8253 | 37.32 |
| 0.125 | 0.0625 | 1024 | 1022 | 23.89 | 5.4 | 0.8318 | 36.99 |
| 0.125 | 0.25 | 16 | 784 | 50.32 | 5.5 | 0.4867 | 33.89 |
| 0.125 | 0.25 | 128 | 879 | 45.24 | 5.4 | 0.5590 | 35.11 |
| 0.125 | 0.25 | 1024 | 994 | 47.24 | 5.4 | 0.5954 | 36.69 |
| 0.125 | 0.5 | 16 | 529 | 100.22 | 5.5 | 0.0551 | 26.93 |
| 0.125 | 0.5 | 128 | 639 | 72.93 | 5.4 | 0.1461 | 29.03 |
| 0.125 | 0.5 | 1024 | 913 | 75.42 | 5.4 | 0.1834 | 34.83 |
| 0.125 | 1 | 16 | 16 | 227.34 | 5.5 | 0.0104 | 2.97 |
| 0.125 | 1 | 32 | 32 | 160.01 | 5.4 | 0.0174 | 3.40 |
| 0.125 | 1 | 64 | 64 | 131.22 | 5.4 | 0.0259 | 4.39 |
| 0.125 | 1 | 128 | 128 | 118.04 | 5.4 | 0.0299 | 6.40 |
| 0.125 | 1 | 256 | 251 | 113.92 | 5.4 | 0.0337 | 10.51 |
| 0.125 | 1 | 1024 | 646 | 115.17 | 5.4 | 0.0353 | 24.19 |
| 0.125 | 1 | 4096 | 906 | 118.44 | 5.4 | 0.0348 | 33.33 |

Table 12: Gen. PPL (↓), entropies and MAUVE (↑) of samples by MDLM trained for 250K steps on OWT.

| $T$ | NFE | Gen. PPL (↓) | Entropy | MAUVE (↑) | Sampling Time (sec) (↓) |
|---|---|---|---|---|---|
| 8 | 8 | 246.70 | 5.6 | 0.0134 | 7.19 |
| 16 | 16 | 109.70 | 5.5 | 0.1353 | 13.81 |
| 32 | 32 | 67.44 | 5.5 | 0.4195 | 27.10 |
| 48 | 48 | 55.96 | 5.5 | 0.5062 | 39.42 |
| 64 | 64 | 51.11 | 5.4 | 0.6123 | 53.48 |
| 128 | 128 | 43.58 | 5.4 | 0.6477 | 106.96 |
| 256 | 251 | 40.44 | 5.4 | 0.6924 | 213.92 |
| 1024 | 657 | 37.15 | 5.3 | 0.7267 | 566.19 |
| 4096 | 907 | 36.48 | 5.3 | 0.7026 | 752.06 |

Table 13: Gen. PPL (↓), entropies and MAUVE (↑) of samples by BD3-LMs trained for 250K steps on OWT.

| Block size | $T$ | $T'$ | NFE | Gen. PPL (↓) | Entropy | MAUVE (↑) | Sampling Time (sec) (↓) |
|---|---|---|---|---|---|---|---|
| 4 | 256 | 1 | 512 | 184.86 | 4.00 | 0.0048 | 26.26 |
| 4 | 512 | 2 | 740 | 216.73 | 4.81 | 0.0081 | 37.44 |
| 4 | 1024 | 4 | 968 | 110.22 | 5.14 | 0.0533 | 49.20 |
| 4 | 2048 | 8 | 1124 | 51.92 | 5.22 | 0.3515 | 56.77 |
| 4 | 4096 | 16 | 1180 | 34.93 | 5.24 | 0.6726 | 60.32 |
| 8 | 256 | 2 | 383 | 267.26 | 4.69 | 0.0061 | 20.58 |
| 8 | 512 | 4 | 584 | 170.50 | 5.04 | 0.0168 | 31.44 |
| 8 | 1024 | 8 | 812 | 80.31 | 5.20 | 0.1479 | 42.14 |
| 8 | 2048 | 16 | 951 | 47.16 | 5.22 | 0.5723 | 50.01 |
| 8 | 4096 | 32 | 1051 | 36.34 | 5.25 | 0.6807 | 55.53 |
| 16 | 256 | 4 | 316 | 240.20 | 5.10 | 0.0114 | 19.36 |
| 16 | 512 | 8 | 515 | 112.56 | 5.28 | 0.0971 | 31.17 |
| 16 | 1024 | 16 | 703 | 61.82 | 5.30 | 0.4067 | 43.76 |
| 16 | 2048 | 32 | 881 | 44.06 | 5.29 | 0.6383 | 53.79 |
| 16 | 4096 | 64 | 984 | 37.61 | 5.29 | 0.7248 | 58.82 |

E.9   EXAMPLE GENERATED SAMPLES BY MODELS TRAINED ON OWT

to be known to the grand jury yet, but it has been explained he could not immediately cause any damage to happen, such as preventing a clean break from someone hacked or creating a fake email. (And again, Hillary's tweet never caused the genesis of the controversy as it was announced, his tweeting violation could easily have changed the course of the matter.)

The Times:

...Senator John McCain doesn's State of the Union...should really have to decide—mossipally—whether they believe to allow a Trump presidency in the first place. There is no situation in which Hillary's campaign could choose to take the matter in a different light.

Except for just one thing what Hillary did in her son's law book there was her "crook of mess" notion.

At this, it is irrelevant today to ask John Podesta to choose someone in Congress so it will be up until the election year, to solve the problems through this simple conceptual framework, which is simple, soft and unhinged and abstract, to create an all too common threadbare" solution.

As an excuse to say, we're okay with the recent DOJ's somewhat unusual way of saying only what the rest of us are thinking in the know.

They knew...the Democratic people of this country set up the proper system to identify.

The legal partner of the campaign and FBI are working with the federal investigation into the Trump campaign for violations of campaign laws under V.W. and Harry Truman.

A joint team star Michael Burnett was allegedly killed after a dog survived a shooting attack by a suspect when cops showed up for a Texas sheriff dog in an afternoon raid on a joint squad and a Texas Border Patrol agent with the animal owner of the state filed charges against Sheriff Edell, Fox and AP reports.Police had been conducting an eight-hour search in order to find the dog dead sometime Monday, during the time of the 100th anniversary of the Golden Gabriel Shooting Act.That was when the Bureau of Investigation allowed the police to close the area after a group of dogs were called to the events, they were, at that time they were found dead.The authorities pulled more than 20 pick-up dogs but were released. Sheriff Edell insisted on using the dogs, given to sheriff's deputies as "an excellent dog.""I'm going further," to deputies and reporters, the sheriff said officers had pulled on the rear door of a drug smuggler and a baggie, which were immediately spotted by private security cameras at the scene.A cat had reportedly appeared on a front door in front of a television screen inside the house in the shooting, Dina Sootoot, who plays...Shanna and A Prairie Winage, were booked for a movie position in the U.S, with a movie star movie and a party dog in their midst.She formerly played Z.A.. During a hour-long episode, on the Texas Weill, he admitted during the interrogation that Mr. Jupp suffered from dramatic seizures that were preceded by a rash.The animal's owner, a doctor, confirmed at the scene that he was overdosed to the illegal drug, a week later was later charged with administering Billing Aid Services. Upon returning to the scene, Fox reported, Mr. Jupp sustained only minor injuries while Mr. Jupp subsequently passed away.Having later moved from Middle Tennessee to South Florida, Mr. Jupp moved to Florida in 2007 on a contractual basis (and with a Green Bay film) and this ultimately landed him in solitary confinement three weeks in a drug row in the desert. Advertisement

"There is a meaningful escape, zero suffering. Repeat Five, jail! Repeat Five Corners!" -and-Healthy physical health Bill (Public

Domain via Getty Images, May17, 2015) Much of the more recently named London Department of Public Buildings Embley

(Flea) made a new investment in approximately $5 Million with the acquisition of a single new office unit comprised of parking spaces and a new 1.6-store five-story studio at the corner of its current office in Coho, London, as part of a three-store-off luxury brick-and-mortar store and several hundred multi-unit studio units, which also include the new airport, under-construction office, reports [LinkedIn.com](http://linkedin.com/) The office is conveniently situated in a building "just over a shopping plaza" and has been "asked for purchase by city officials but not to allow it there one could use."

Figure 21: An unconditional sample ($L = 1024$) from Eso-LM ($\alpha_0^{\text{train}} = 1$) trained for 250K on OWT using inference-time hyperparameters $\alpha_0^{\text{eval}} = 1$ and $T = 1024$. This corresponds to an NFE of about 646 and a sampling time of 24.19 seconds per batch of 512 samples. Gen. PPL, entropy and MAUVE are 49.05, 5.3 and 0.4939 respectively.

and for much of its population, Auckland is still of significant interest to both companies.

The public can also afford to copy companies such as Gotham, with offices in New York suburbs such as New York, followed by larger commercial spaces such as London's Empire Bridge and Gotham.

Small Business; but have office space in Auckland; expertise perfect for marketing results.

- Startup advertising work. Put on billboards such as National Grid are ideal for digital marketing work. A flat screen television that got the mind-set

5 hours-by-hour traffic must be in television advertising

The Michaelarinen Gates Shayka-Tin did with his first down in marketing was to Compromise your business, very easy to do.

As the pressure from you surrounds it with work and you're quite healthy, it is still possible to invest just a few dollars a month — your salary or whatever, the money chosen to share the press — via a marketing campaign with FreeMedia.

He said she used to think that the modern internet was paramount: "Follow not one of the most popular people in the world. If they are 50, find a way to have two kids their age. Or, if they are a celebrity, too. The same applies very well, television has that.

It's a way, at least in my opinion, to connect yourself and others and if you sell yourself a bit of confidence.

Read more:

"Can you afford an online lifestyle where you don't know it? Tell your opinion or credibility through information or speech. If you can, you don't need it all the time."

On the other hand, of course, it's a much better thing, for example, to need to offer up a genuine chance to walk with people looking, on camera, and in a hands-on manner of confidence.

Take all of that approach. "You can also try and narrow down the perspective everything that was natural would be easy, which is true if advertisements are not marketed that way.

When advertising that someone named you said was a television advertisement was, when, think of television, the internet was it - and they have no editorial authority; there's no PR for Free Media, but every advertisement is a commercial of their own.

Is that that true?

Yeah. No. Because you've worked in advertising for a very long, maybe for a while. They worked and made friends with their jobs today and you still haven't thought about it at all.

It is a world at best.

For me, from the newspapers, to the advent of the internet, I was constantly looking to appeal to the "new people" that I always connected with, and everyone loved, Twitter.

But now it is still true.

If you haven't all the young author books. Download our free online video guide for your audience for this expert advice.

Read the full interview: Tom Moss covers hundreds of news outlets in Japan and Australia. His work is for letters and written back millions of times. From riding horses to e-reading devices, ATM machines.

For us their ads for these pages already take up more than 1.5 viewers and 30 hours a week. The opportunity to read things and bring you more.

"The internet is never digital for everybody, I would be thrilled if it's the user I've seen before," he said,: "The reality is there is this new age for business is that you're the best as you possibly can and have a feeling they deserve it.

Don't look for cheap TV, and no business editor should pay attention to it.<|endoftext|>In a 2017 television news magazine interview, newly-minted investor Warren Buffett noted that the top income level was increasing at approximately half that amount, but the 2016 American economy "has been operating at a level that most thought would have been a bubble burst."

Buffett said that those years or so, an average American has been earning almost 40 percent in the last quarter, including this for the past five years. That is why, as traditional high earners, businesses must make enormous gains in income tax're worth about 20 percent of their CEO's income. Even those high earners make more.

Advertisement

Advertisement

In the beginning to end, although most sports today make the earnings for all Americans, in the past decades have provided the entertainment revenue, especially at the home entertainment market. Most people have very little disposable income — jobs, living games and using for free. That's their source of income, but they don't provide nearly enough information. So a news article is entitled, "Why Americans are working too hard and don't make more."

Advertisement

Here's the American experiment

Figure 22: An unconditional sample ($L = 1024$) from Eso-LM ($\alpha_0^{\text{train}} = 1$) trained for 250K on OWT using inference-time hyperparameters $\alpha_0^{\text{eval}} = 1$ and $T = 64$. This corresponds to an NFE of about 64 and a sampling time of 4.39 seconds per batch of 512 samples. Gen. PPL, entropy and MAUVE are 61.43, 5.4 and 0.4166 respectively.

the modern Thecat race over where this may turn and welcome themselves with their futuristic agility. However, the could be and possibly not at all that backed up. In mentioned, I think the major key issues is balance, ie perhaps the best weapon is a right handed side. While balance - any - always has a presence, a lot of things should never stay like the spine and lean to both legs. Whilst it how wide and, you can also swing wide this making it impossible for a pinch bat guard without weapons. With contrast, the With more than one side, there will be more options than the if it, but allow the the most difficult primary weapon of being in any and balancing out the balanced side. For example, the best players need sharp but when the backup b bat side might be stiff and this be easy. you could swing back then-trod right bat side and a double-beast it and that would work. There for me is a smart side but weird bat side does not bats well So that is always a balance, the bats may not like it but they always might be with one side anyway. bob is skills are learnt and if every bat, has a try out and wrong side to manage to even in and out of the bat. Work to make it and when easy. this is perhaps another issue. to have able to bat in whatever the wrong side is required for a bat that would always last and can always develop into a game especially though trying to have met your bat a bit before is also an issue. With a batter knows their T bat regularly, occasionally you might even pick wiff bat which just means no. I know that it worked but when I had. first try duff bat regularly and return to how they more or less. good

L :There it doesnt seem to work and said it doesn't work the way you want to do it would also work. It showed you had a nice batting set or secondary bat side and would be be great anders to trouble guys with good tiered shots and can I say this from a y bat perspective as I and have both feel as to some level of smart bat. Most of the time, however, I don't think they are a very good bat. they are novice batters and sometimes not the only good bat for even the best right foot bat. On today's point of course, they just have to be third first or second second defensive often on the bat left side, the bat right bat side or on the end of the bat, and have a couple of hands on used to holding the bat bat to the other side of the bat. bat bat is very powerful.

L :So it is working well at best, there is still a little bit of ability to park your bat as expected, but bat won't work with to base error bats and hitting some or-side could still possible. How do you decide to just start the third bats which would make the bat look effective while not very will be one for respond, or R :In a smaller group of slower bat hitters particularly bats u a it is not very weak bat they will think they are playing better with bat than short bat, bat has already developed in terms of bat learning but I do not believe that the bat learned

L : If you are doing bantops, I have people not trying to learn anything. hassleds's bat learning. you should always learn bantops.

L : Well bantops is bat or Obleto bat is bat can get you really into a bat training box instead of being it being training box or be described as a bat session at the light of baters what.

L : They are easy to understand bat training designed bats. ly designing bats are not so and useful but maybe they are better, one being able to bat right hand in right hand defend left left bat bat bat is than batting left hook bat bat is than holding bat bat. at least this difference has started to play out recently for myself. play time between defensive and offensive bat, the do of said bat bat is near when he stole bat from him. but they bat the ball from bat to to bat bat. against bat bat position too bats like that, you have attack average bat with short bat. you're going to catch the bat very low there and still with ball kick into bat bat. in certain situations, when a bat bat can be dealt, sometimes. on the end of the bat, maybe third bat, another bat which is third bat, so if bat bats at third bat and the second bat a second bat. then they go to a third bat or hold second bat. they bat handle it better. you can take bat to third second main bat. end of the bat so then bat to your main bat from where bat go second bat. bat, second bat. bat, the bat, on deck. double bats, extra bat, always with bat and bat. no extra bat. less bat bat. A little extra bats"

Figure 23: An unconditional sample ($L = 1024$) from BD3-LM ($L' = 4$) trained for 250K on OWT using inference-time hyperparameter $T = 256$ ($T' = 1$). This corresponds to an NFE of about 512 and a sampling time of 26.26 seconds per batch of 512 samples. Gen. PPL, entropy and MAUVE are 184.86, 4.0 and 0.0048 respectively. Note that this sample appears incoherent compared to those with similar sampling time from Eso-LMs.

## APPENDIX F    THE USE OF LARGE LANGUAGE MODELS

We used LLMs in paper writing to identify grammar mistakes.

