# OpenReview forum: "Esoteric Language Models"
_ICLR.cc/2026/Conference — Submitted to ICLR 2026_

### Official Review · Reviewer_hFRL · 2025-10-31

**Soundness:** 3
**Presentation:** 3
**Contribution:** 2
**Rating:** 6
**Confidence:** 3

**Summary:**

The most important contribution of this paper is a novel hybrid architecture that combines the AR and MDM paradigms for text generation. This method can be viewed as a smooth interpolation between AR and MDM, and shows the potential for a trade-off of their pros and cons.
Another interesting contribution is the KV-caching scheme for the AR component during inference, which can improve sampling efficiency.

**Strengths:**

- The presentation of this paper is clear and easy to follow.
- The proposed hybrid architecture that combines the AR and MDM paradigms introduced a flexible dimension for model design space, i.e., $\alpha_0$. The experiment in Section 5.2 suggests that a non-trivial choice of $\alpha_0$ makes sense.
- The proposed KV-caching scheme for the AR component during inference can improve sampling efficiency.
- The

**Weaknesses:**

- The authors put their emphasis on the trade-off between efficiency and speed. However, many studies also pay attention to the global understanding ability and better generalization on downstream tasks of diffusion models. I think the authors should provide more evidence on these aspects.
- I think the sentence "This is the first time for IW bounds to be obtained for discrete diffusion" in Line 393 overclaims their contribution.
- In Figure 2, the curve exhibits a sudden decrease when $\alpha=0.125$. Does this imply that this method is not robust enough?
- MDM generally samples slower than AR, as KV caching is not enabled. Also, AR tends to give a lower perplexity. Then how can we expect a trade-off between efficiency and speed?

**Questions:**

- In line 425, the authors state that settign $\alpha^{train]=1$ performs the best. Doesn't this seem contradictory with the motivation of interpolating between AR and MDM?

---

> ### Author Response · Authors · 2025-11-20
> **Response (1/5) to Reviewer hFRL**
>
> We appreciate the reviewer’s recognition that our paper is clearly presented and introduces a novel and effective method for addressing the quality–efficiency tension between AR and diffusion LLMs.
>
> We now address the reviewer’s comments and questions below.
>
> # Concern 1: How can you interpolate with MDM which is slower and doesn’t have KV caching?
>
> Excellent question. We clarify below:
>
> > MDM generally samples slower than AR, as KV caching is not enabled.
> >
>
> Correct. As a first step, we improve MDM sampling efficiency by replacing its bidirectional attention with causal attention applied to a shuffled input sequence `[lines 243–251 & Sec 4.2]`. This enables KV caching and makes **our Eso-LMs on par with or faster than AR models in terms of speed, though with slightly lower sample quality**.
>
> > Also, AR tends to give a lower perplexity.
> >
>
> True. AR models produce higher-quality samples, **motivating our hybrid approach that interpolates between AR and MDM behaviors**.
>
> > Then how can we expect a trade-off between efficiency and speed?
> >
>
> Since Eso-LMs are on par with or faster than AR models, sample quality is adjusted via $\alpha_0$, which controls the proportion of tokens generated in diffusion versus autoregressive mode. Refer `Fig. 1` for a visual illustration.

---

> ### Author Response · Authors · 2025-11-20
> **Response (2/5) to Reviewer hFRL**
>
> # Concern 2: IW bounds are not a big contribution
>
> We respectfully disagree. Likelihood or perplexity is a central evaluation metric for LLMs. A key critique of comparing MDMs and AR models via perplexity is that for MDMs it represents a bound on the likelihood, whereas for AR it is exact. Additionally, in `Table 2`, it's highly useful to know if the relative order of perplexities (ELBO) for Eso-LMs align with the relative order of their true exact perplexities. Thus, enabling exact likelihood computation for diffusion LLMs is a valuable contribution.
>
> **Challenges:**  IW bounds have been used to estimate exact likelihoods for models like VAEs [1], but diffusion models pose a challenge due to infinitely many latent variables. For Gaussian diffusion [2], the exact likelihood relies on PF-ODEs, which are unavailable for discrete diffusion.
>
> **Our Contribution:**  We introduce the **first formula for computing the exact likelihood of Masked Diffusion Models**, applicable to both MDLM and Eso-LMs:
>
> $-\log p(x) = \lim_{K \to \infty} - E_{\sigma_1,\ldots,\sigma_K} \left[ \log \frac{1}{K} + \log \sum_{k=1}^K \exp\left({\sum_{l=1}^L \log p_\theta (x_{\sigma_k(l)} \mid x_{\sigma_k(<l)})}\right) \right]$
>
> where $\sigma_k$ denotes a permutation of the clean sentence $\mathbf{x}$.
>
> **Computing the exact likelihood is possible due to our two key insights:**
>
> 1. Expressing the IW bounds using the Any-Order objective.
> 2. Leveraging our proposed Eso-LMs, which uniquely allow efficient evaluation of this formula.
>
> ---
>
> **References**
>
> [1] Burda et al, 2015, "Importance weighted autoencoders."
>
> [2] Song et al., 2021, "Score-based generative modeling through stochastic differential equation."

---

> ### Author Response · Authors · 2025-11-20
> **Response (3/5) to Reviewer hFRL**
>
> # Concern 3: Downstream tasks performance
>
> > many studies also pay attention to the global understanding ability and better generalization on downstream tasks of diffusion models. I think the authors should provide more evidence on these aspects.
> >
>
> Our work, like prior studies [1, 2, 3], focuses on unconditional text generation as the primary downstream task. We evaluate using multiple metrics, including Generative Perplexity and Mauve [4], and observe that **Eso-LMs effectively interpolate between the sample quality of diffusion and autoregressive models while being significantly faster than the baselines** [1, 3].
>
> Scaling these models to tasks such as coding and reasoning is an important next step. However, given our limited computational resources as a small academic lab, we leave this direction for future work.
>
> ---
>
> **References**
>
> [1] Sahoo et al., 2024 "Simple and effective masked diffusion language models."
>
> [2] Sahoo et al., 2025 “The diffusion duality.”
>
> [3] Arriola et al., 2025 "Block Diffusion: Interpolating Between Autoregressive and Diffusion Language Models."
>
> [4] Pillutla et al., 2021 “MAUVE: Measuring the Gap Between Neural Text and Human Text using Divergence Frontiers.”

---

> ### Author Response · Authors · 2025-11-20
> **Response (4/5) to Reviewer hFRL**
>
> # Clarification 1: Sudden decrease in Figure 2
>
> In `Fig. 3`,  we report the Pareto frontier of Gen PPL (Generative Perplexity) across sampling steps/time, showing the best Gen PPL among models trained with $\alpha^\text{train}_0 \in \\{1, 0.5, 0.25, 0.125\\}$ and evaluated with $\alpha^\text{eval}_0 \in \\{1, 0.5, 0.25, 0.0625\\}$ as described in `Lines [457–458]`.
>
> While a noticeable kink appears in the Gen PPL frontier for Eso-LMs, this effect is much weaker for MAUVE (`Fig. 4`). We do not yet have a definitive explanation for this behavior.

---

> ### Author Response · Authors · 2025-11-20
> **Response (5/5) to Reviewer hFRL**
>
> # Clarification 2: Best $\alpha^\text{train}_0$
>
> > The authors state that setting $\alpha_0^{\text{train}} = 1$ performs the best. Doesn't this seem contradictory with the motivation of interpolating between AR and MDM?
> >
>
> This is a **misunderstanding**. Our point was that varying $\alpha_0$ during evaluation (via multiple different $\alpha^\text{eval}_0$) allows a model trained with only $\alpha^\text{train}_0 = 1$ to **nearly match** the Pareto frontier achieved by multiple models trained with $\alpha^\text{train}_0 \in \\{1, 0.5, 0.25, 0.125\\}$. **In other words, interpolation at inference time is generally sufficient.**
>
> As shown in `Fig. 16`, the model trained with $\alpha^\text{train}_0 = 1$ achieves a best MAUVE of  $\approx 0.7$, while the best overall MAUVE (≈ 0.8) is obtained for $\alpha^\text{train}_0 = 0.125$. Therefore, **tuning $\alpha^\text{train}_0$ can help squeeze out better performance.**

---

### Official Review · Reviewer_naX2 · 2025-11-01

**Soundness:** 3
**Presentation:** 2
**Contribution:** 2
**Rating:** 4
**Confidence:** 5

**Summary:**

The paper proposes Eso-LMs, a hybrid language modeling framework that interpolates between masked diffusion models (MDMs) and autoregressive (AR) models. The core ideas are: (i) a two-phase sampling procedure where some tokens are denoised in parallel (diffusion) and the rest left-to-right (AR), and (ii) an attention-bias scheme that enables causal attention and unified KV caching even during diffusion, improving inference efficiency. A variational bound decomposes training into an AR loss over masked positions plus an MDM loss, with a hyperparameter controlling the interpolation. Empirically, Eso-LMs achieve competitive perplexities versus MDMs and a better speed–quality Pareto frontier, with large speedups at long context via KV caching.

**Strengths:**

- Clear hybrid objective with principled ELBO derivation and interpolation knob $\alpha_0$ (Sec. 3.1, Eq. 7).
- Practical sampling schedule plus attention bias enabling KV caching in both phases; addresses a key MDM bottleneck (Sec. 3.2, 4.2).
- Strong speed–quality trade-offs and long-context latency improvements vs. MDLM/BD3-LM (Figs. 3–4; Table 9).
- Useful analysis of importance-weighted bounds for discrete diffusion to approximate true PPL (Sec. 3.3; Table 2, Table 6)

**Weaknesses:**

- Sampling schedule clarity: Sec. 3.2 introduces a “Denoising Schedule” by name before defining how it is computed; the detailed procedure is deferred to Appx. B.3 and could be surfaced earlier for readability [p4, lines 212–215].

- Efficiency restriction explanation: The claim “restrict the forward pass at step k to only the previously denoised tokens and the current mask tokens” would benefit from a precise equation/attention mask description on the main text page [p4, lines 223–229].

- Diffusion-phase attention assumption: Sec. 4.1.1 states “mask tokens are denoised using only clean tokens but clean tokens do not attend to mask tokens,” which deviates from common MDM training with bidirectional attention; the rationale relies on the proposed causal-bias construction and AO-ARM connections, but the text could more explicitly contrast with standard MDM [p5, lines 262–267].

- Train–test mismatch concerns: The sequential-phase uses a concatenation $z_0\oplus x$ for training (Sec. 4.1.2), but this concatenation is “unnecessary” at sampling; please justify that this does not introduce a distribution shift or extra leakage [p5–6, line 287].

**Questions:**

- Sec. 3.2, line 212: How exactly do you pre-compute the order in which tokens are denoised? Please make it clear in the main text with a concise algorithm.

- Sec. 3.2, line 223: “Restricting the forward pass at step k to only the previously denoised tokens” — could you provide a compact equation for the attention bias or a figure on the main page that shows Q/K/V subsets and KV reuse at step k?

- Sec. 4.1.1, line 262: The statement “mask tokens are denoised using only clean tokens but clean tokens do not attend to mask tokens” is unclear. Please reconcile this with standard MDM training where mask and clean tokens interact bidirectionally, as in Large Language Diffusion Models (Nie et al., 2025). What assumption or theoretical result justifies your restriction?

- Sec. 4.1.2, line 287: The training concatenation $z_0\oplus x$  seems different from inference. Is there a gap between training and inference? Please specify conditions under which this is unbiased and show an ablation that removes concatenation.

---

> ### Author Response · Authors · 2025-11-18
> **Response (1/4) to Reviewer naX2**
>
> We appreciate the reviewer’s recognition that our hybrid language modeling framework is mathematically principled, computationally efficient during inference, and effective in improving the speed-quality trade-off over relevant baselines.
>
> We now address the reviewer’s comments and questions below. All added contents in the submitted PDF are shown in orange.
>
> # Clarification 1: Pre-computing the Denoising Order during inference
>
> > Sec. 3.2, line 212: How exactly do you pre-compute the order in which tokens are denoised? Please make it clear in the main text with a concise algorithm.
> >
>
> We kindly note that **`Sec. 3.1`** already provides a detailed description of the denoising order, which consists of two phases:
>
> 1. **Diffusion phase:** Tokens are denoised in a **random order**, and the number of tokens denoised per step depends on the selected number of denoising steps. The fraction of tokens denoised in each mode is controlled by the hyperparameter $\alpha_0$, such that an $\alpha_0$ fraction is denoised in the diffusion phase.
> 2. **Autoregressive phase:** While the remaining $(1 - \alpha_0)$ fraction is denoised in the autoregressive mode where all the mask tokens are filled in a left-to-right manner, one token at a time.
>
> As requested by the reviewer, we have included the algorithm to compute the denoising schedule in **`Alg. 1`**.
>
> > Sec. 3.2 introduces a “Denoising Schedule” by name before defining how it is computed; the detailed procedure is deferred to Appx. B.3 and could be surfaced earlier for readability [p4, lines 212–215]
> >
>
> We agree with the reviewer’s feedback that, to further clarify things, we should move the algorithm for pre-computing the denoising order (Appendix B.3) from the Appendix to the main text. **We have made these changes in orange in the revised PDF.**

---

> ### Author Response · Authors · 2025-11-18
> **Response (2/4) to Reviewer naX2**
>
> # Clarification 2: “Restrict the forward pass at step k to only the previously denoised tokens and the current mask tokens” — needs equation/figure in the main text
>
> > Sec. 3.2, line 223: “Restricting the forward pass at step k to only the previously denoised tokens” — could you provide a compact equation for the attention bias or a figure on the main page that shows Q/K/V subsets and KV reuse at step k?
> >
>
> We kindly note that `Fig. 1` already illustrates how “KV reuse” of Eso-LMs work on a concrete example (sequence length = 8).
>
> We acknowledge the reviewer’s feedback that a compact mathematical equation for the attention bias could be helpful; **we have now moved this equation from `Appendix B.6` to `Sec 4.2` in the revised PDF**.

---

> ### Author Response · Authors · 2025-11-18
> **Response (3/4) to Reviewer naX2**
>
> # Concern 1: The attention mechanism of our method should be contrasted more explicitly with that of MDLM
>
> > Sec. 4.1.1 states “mask tokens are denoised using only clean tokens but clean tokens do not attend to mask tokens,” which deviates from common MDM training with bidirectional attention; the rationale relies on the proposed causal-bias construction and AO-ARM connections, but the text could more explicitly contrast with standard MDM [p5, lines 262–267].
> >
>
> > Sec. 4.1.1, line 262: The statement “mask tokens are denoised using only clean tokens but clean tokens do not attend to mask tokens” is unclear. Please reconcile this with standard MDM training where mask and clean tokens interact bidirectionally, as in Large Language Diffusion Models (Nie et al., 2025). What assumption or theoretical result justifies your restriction?
> >
>
> **Background**: In Masked Diffusion Models (MDMs), given a noisy sequence  $\mathbf{z}_t$ containing mask tokens, the denoising transformer is trained to fill in these masked tokens. This setup is closely related to **Masked Language Models (MLMs)** such as BERT [1], where a **bidirectional transformer** predicts masked tokens from surrounding context. Consequently, prior works such as MDLM, Sedd, LLada [2, 3, 4] have typically adopted **bidirectional transformers** as the denoising backbone for MDMs.
>
> **Challenges with the current approaches:** As described in `[Lines 243–251]`, bidirectional attention leads to **severe inference inefficiency**:
>
> 1. the activations corresponding to the clean tokens change during inference which prevents KV caching.
> 2. Perform forward pass of the noisy sequence containing all mask tokens even though only a subset of those are denoised
>
> **Rationale behind our approach:**   These limitations motivate our design. As discussed in `[Lines 243–251 and 304-310]`, an **ideal inference scheme** should:
>
> 1. Perform forward passes **only for the subset of mask tokens** selected for denoising (as defined in Algorithm 1).
> 2. Keep activations of **clean tokens fixed** throughout inference.
> 3. Ensure each mask token depends **only on clean tokens**, not on future tokens to be denoised.
> 4. **Denoise tokens in random order** to maintain stochasticity.
>
> We achieve this efficiently by **shuffling the input sequence** $\mathbf{z}_t$ so that **clean tokens are grouped on the left** and **mask tokens on the right**, followed by applying a **causal attention mask**.
>
> A clear visualization of this procedure is provided in **`Fig. 1`** of the main paper and detailed in `Sec 4.2`.
>
> ---
>
> **References**
>
> [1] Devlin et al., 2018 “BERT: Pre-training of Deep Bidirectional Transformers for Language Understanding”
>
> [2] Sahoo et al., 2024 "Simple and effective masked diffusion language models."
>
> [3] Lou et al. 2024 “Discrete diffusion modeling by estimating the ratios of the data distribution”
>
> [4] Nie et al. 2025 “Large Language Diffusion Models"

---

> ### Author Response · Authors · 2025-11-18
> **Response (4/4) to Reviewer naX2**
>
> # Concern 2: Training / Inference mismatch
>
> > The sequential-phase uses a concatenation of z0 and x for training (Sec. 4.1.2), but this concatenation is “unnecessary” at sampling; please justify that this does not introduce a distribution shift or extra leakage [p5–6, line 287].
> >
>
> This **concatenation is applied only during training**, not during inference. Below we provide a concrete example to illustrate why this technique is necessary for the sequential-phase training of Eso-LMs and why it **does not introduce any distribution shift** between training and sampling.
>
> Given:
>
> - Clean sequence ($\mathbf{x}$): `ABCDEFGH`,
> - Randomly masked sequence ($\mathbf{z}_0$): `AB[M1]DE[M2]GH` (`[M1]` and `[M2]` represent mask tokens)
>
> During the sequential training phase, the denoising model needs to predict:
>
> 1. `C` for `[M1]`
> 2. `F` for `[M2]`,  assuming `C` exists at `[M1]`. This step is very challenging because this contradicts the above (step 1).  To resolve this, we perform a forward pass of the concatenated sequence  `ABCDEFGH AB[M1]DE[M2]GH` where `[M2]` attends to `C` instead of `[M1]` . In general, this ensures that  `[M_i]` attends to the clean tokens corresponding to all  `[M_{j < i}]` .
>
> During sampling, this conflict is naturally resolved because all the mask tokens are filled by clean tokens auto-regressively from left to right. A visual depiction is provided in `Fig. 1`.
>
> **Comparison to BD3-LMs** [1]: Our main baseline, BD3-LMs, also interpolates between AR and diffusion and follows a similar concatenation approach. However, BD3-LMs **double the sequence length for the entire batch**, making them **2.67× slower than MDLM** [2].  In contrast, **Eso-LMs are ≈2× faster to train than BD3-LMs** while outperforming them on speed-quality Pareto frontier; see `Figs. 3, 4`.
>
> ---
>
> **References**
>
> [1] Arriola et al., 2025 "Block Diffusion: Interpolating Between Autoregressive and Diffusion Language Models."
>
> [2] Sahoo et al., 2024 "Simple and effective masked diffusion language models."

---

### Official Review · Reviewer_RnKE · 2025-11-01

**Soundness:** 3
**Presentation:** 2
**Contribution:** 3
**Rating:** 6
**Confidence:** 3

**Summary:**

This paper introduces a hybrid LLM paradigm that aims to unify autoregressive (AR) and masked diffusion model (MDM) for language modeling. The key idea is to interpolate between AR and MDM objectives. Correspondingly, the transformer architecture is modified to accommodate both causal (AR) and bidirectional (MDM) attention, and — crucially — enable KV caching during diffusion.
This design allows Eso-LMs to achieve parallel generation while retaining AR-level inference speed.
Experiments on LM1B and OpenWebText show that Eso-LMs interpolate between diffusion and AR perplexities, establish better results on the speed–quality Pareto frontier.

**Strengths:**

* Timely and relevant topic: Bridging the gap between AR and diffusion models addresses a central limitation in current LLMs, shedding light to how to better balance quality and efficiency in language modeling.

* Clear technical contributions: The proposed hybrid formulation and unified attention mechanism are clearly presented and novel to me. The derivations seem technically sound.

* Strong empirical results: The model demonstrates consistent improvements over both MDM and block-diffusion baselines in perplexity and inference efficiency on benchmarks. Even though the scale is relatively small, the results look promising.

**Weaknesses:**

### Presentation:
Even though the high-level idea is clearly presented in Sec 3.1, there are a lot of technical details in the implementation process and I feel these subsequent designs are not well-justified and discussed. The ablation analysis is somewhat limited given the number of components involved.

### Scientific value
I appreciate this work on the technical level very much. But I think the scientific values can be further improved.
The combination of AR and diffusion objectives feels somewhat mechanical—focused on unifying two training losses rather than addressing a specific modeling limitation or data property.
It remains unclear what type of data/scenario actually benefits from this interpolation, e.g., general, coding, math, agent, etc.
Given that actually implementing the proposed model requires significant changes to the training/inference pipelines and transformer architectures, and that setting alpha_0 to 1 or 0 performs significantly worse than standard MDM and GPT, I feel this paper could provide more intuition/justification for why this interpolation helps beyond the observed empirical trade-offs in limited benchmarks.

**Questions:**

* The decomposition in Eqn (5) seems a bit artificial. Could you provide more intuitive motivation for why this hybrid formulation could yield a better generative model, and under what data regimes (some sufficient conditions) it is expected to help?

* In table 5 of appendix C.1, what is the reason for the divergence?

* How does this work relate to the any-order GPT line of work [1,2], which also kind of combines GPT and MDM?

* If my understanding is correct, for different values of \alpha_0, we have to re-train the model from scratch. Is there some potential methods that can accommodate multiple \alpha_0's in one training run?

[1] σ-GPTs: A New Approach to Autoregressive Models, arxiv 2404.09562

[2] Any-Order GPT as Masked Diffusion Model: Decoupling Formulation and Architecture, arxiv 2506.19935

---

> ### Author Response · Authors · 2025-11-21
> **Response (1/6) to Reviewer RnKE**
>
> We appreciate the reviewer’s recognition that our method is novel, technically sound, and effective in addressing the quality-efficiency tension between AR and Diffusion LLMs.
>
> We now address the reviewer’s comments and questions below.
>
> # Concern 1: Eso-LMs require significant changes to the training / inference pipelines and transformer architectures
>
> > implementing the proposed model requires significant changes to the training/inference pipelines and transformer architectures […]
> >
>
> We respectfully disagree with this characterization. **The modifications introduced by Eso-LMs relative to MDLMs are minimal.**
>
> **Training:** We shuffle the input sequence and apply a causal mask for diffusion (`Lines [307–310]`), then concatenate the noisy and clean sequences while constructing an attention mask for the AR phase (`Lines [341–343]`). **These steps can be implemented in just a dozen lines of Python;** see PyTorch code in `Figure 11` . Although `Eqs. 10–16` may appear complex, they are included only for rigor and completeness, as noted in `Lines [341–343]`.
>
> **Inference:** As shown in `Fig. 1`, our inference closely mirrors that of standard autoregressive models, both using causal attention. The only distinction is that AR models predict the next token directly, while Eso-LMs predict the set of next tokens from a set of mask tokens.

---

> ### Author Response · Authors · 2025-11-21
> **Response (2/6) to Reviewer RnKE**
>
> # Concern 2: Technical details not justified
>
> > Even though the high-level idea is clearly presented in Sec 3.1, there are a lot of technical details in the implementation process and I feel these subsequent designs are not well-justified and discussed.
> >
>
> We respectfully disagree with this comment. We provide the justification for our training recipe:
>
> 1. **Diffusion Phase**
>     1. **Proposed changes:** Replace bidirectional attention in MDLM with causal attention over shuffled sequence
>     2. **Justification:** Causal attention uniquely supports KV caching (see `Figure 1`)
>     3. Detailed explanation: Refer to `[Lines 243-251]` and `[Lines 305-310]`, which were also included in the original manuscript
> 2. **Autoregressive Phase**
>     1. **Proposed changes:** Concatenate each randomly masked sequence ($\mathbf{z}_0$) with its corresponding clean sequence ($\mathbf{x}$) during training
>     2. **Justification:** This is to ensure that, when predicting each mask token, the model has access to all clean tokens on its left.
>     3. Detailed explanation: Refer `Lines [314-320]`, which were also included in the original manuscript

---

> ### Author Response · Authors · 2025-11-21
> **Response (3/6) to Reviewer RnKE**
>
> # Concern 3: Limited Ablations
>
> > The ablation analysis is somewhat limited given the number of components involved.
> >
>
> We respectfully **disagree** with this comment. In total, we have trained **34 models** amounting for almost **9000 H200 GPU hours, ablating every new innovation introduced in this paper**. We emphasize that all our experiments are pre-training experiments, which are highly compute intensive.
>
> ---
>
> ## Ablation 1: $\alpha_0$ interpolates between Diffusion and Autoregression
>
> We train **9 models** with different $\alpha_0 \in \\{0.0625, 0.125, 0.25, 0.5, 1.0\\}$ on LM1B and OWT amounting to **2900 H200 hours** and demonstrate that **Eso-LMs smoothly interpolate between the Diffusion and AR perplexities** `Tabs. 1 & 2`.
>
> ---
>
> ## Ablation 2: Attention Mechanisms
>
> For this ablation, we trained **9 models** amounting to **2900 H200 GPU hours**.
>
> As the reviewer correctly points out, Eso-LMs in the full diffusion mode $(\alpha_0 = 1)$ is worse in terms of perplexity than MDLM. There are two steps involved in MDLM → Eso-LMs :
>
> In MDLM, we have bidirectional attention over the entire context. In this paper, we proposed the following two changes:
>
> - Change 1: Apply causal attention only on the mask tokens with bidirectional attention among the clean tokens.
> - Change 2: Apply causal attention among the clean tokens and also the mask tokens
>
> Incorporating these changes into MDLM lead to our methods:
>
> - MDLM + `Change 1` = Eso-LM (A) ($\alpha_0=1$) [Ablation]; described in `Lines [430-448]`
> - MDLM + `Changes 1 and 2` = Eso-LM ($\alpha_0=1$)
>
> Comparing their PPLs on LM1B:
>
> | Method | Clean Tokens | Mask Tokens | PPL (LM1B) |
> | --- | --- | --- | --- |
> | MDLM | Bi-directional | Bi-directional | 31.8 |
> | Eso-LM (A) ($\alpha_0=1$) [Ablation] | Bi-directional | Causal | 31.0 |
> | Eso-LM ($\alpha_0=1$) | Causal | Causal | 35.0 |
>
> **Takeaway:** The difference in PPL between MDLM and Eso-LMs $(\alpha_0 = 1)$ is largely due to the causal attention among the clean tokens since Eso-LM (A) $(\alpha_0 = 1)$ and MDLM have similar perplexities.
>
> Note: PPLs for other $\alpha_0$ values and for OWT can be found in `Tabs. 7 & 8` respectively.
>
> ---
>
> ## Ablation 3: Batch split
>
> For this ablation, we trained **16 models** amounting to **3000 H200 GPU hours**.
>
> The likelihood bound of Eso-LMs is a mixture of Auto-regressive and Diffusion objective. As described in `Lines [298-301]`, a fraction ($\kappa$) of the training examples within a batch are trained with the diffusion objective and the remaining are trained with the AR objective.
>
> A key question is whether the degree of auto-regressiveness in the NELBO (controlled by $\alpha^\text{train}_0$) correlates with the proportion of training samples assigned to the AR objective (controlled by $\kappa$).
>
> Validation PPL ($\downarrow$) on LM1B:
>
> | $\alpha^\text{train}_0$ \ $\kappa$ | 0.75 | 0.5 | 0.25 | 0.125 |
> | --- | --- | --- | --- | --- |
> | 0.5 | 32.3 | **31.5** | Diverged | Diverged |
> | 0.25 | 30.5 | **29.3** | Diverged | Diverged |
> | 0.125 | 27.7 | **26.7** | Diverged | Diverged |
> | 0.0625 | 25.9 | **25.1** | Diverged | Diverged |
>
> **Takeaway:** $\kappa = 0.5$ yields the best performance across a range of $\alpha^\text{train}_0$  values.

---

> ### Author Response · Authors · 2025-11-21
> **Response (4/6) to Reviewer RnKE**
>
> # Concern 4: Setting $\alpha_0$ to 1 or 0 performs significantly worse than standard MDM and GPT
>
> > setting alpha_0 to 1 or 0 performs significantly worse than standard MDM and GPT
> >
>
> ## Eso-LMs ($\alpha_0=1$) vs MDM
>
> The perception that Eso-LMs ($\alpha_0=1$) underperform MDM likely arises from `Tabs. 1, 2`, where Eso-LMs show higher perplexity. As acknowledged in `Lines [422–424]`, this gap is expected.
>
> However, **perplexity is not an ideal metric for comparing Eso-LMs and MDLM**, since it reflects sample quality under an infinite sampling budget and fails to capture performance in realistic finite-time settings. Consequently, it overlooks the **practical advantages** of Eso-LMs.
>
> **Advantage 1: Eso-LMs outperform MDLM on speed-quality Pareto frontier**
>
> 1. With sparse attention, Eso-LMs enable KV caching, making each denoising step significantly faster than MDLM.
> 2. Under a fixed sampling-time budget, Eso-LMs can perform **more denoising steps** and generate **higher-quality samples**, as shown in `Figs. 3, 4`.
>
> **Advantage 2: Eso-LMs support exact likelihood estimation unlike MDLM**
>
> 1. We introduce the first formula for computing exact likelihoods in Masked Diffusion Models, applicable to both MDLM and Eso-LMs:
>
>     $-\log p(x) = \lim_{K \to \infty} - E_{\sigma_1,\ldots,\sigma_K} \left[ \log \frac{1}{K} + \log \sum_{k=1}^K \exp\left({\sum_{l=1}^L \log p_\theta (x_{\sigma_k(l)} \mid x_{\sigma_k(<l)})}\right) \right]$
>
>     where $\sigma_k$ denotes a permutation of the clean sentence $x$.
>
> 2. In practice, $K \approx 100$ suffices to approximate the exact likelihood.
> 3. As discussed in `Lines [261–264]`, **this computation is intractable for MDLM**, since it requires $L (\approx 1000)$ forward passes. In contrast, Eso-LMs can compute it in a single forward pass for a given $K$.
>
> ---
>
> ## Eso-LMs ($\alpha_0=1$) vs AR
>
> We assume that the term “GPT” refers to the “autoregressive models” in our work.
>
> **Perplexity:** As shown in `Tabs 1, 2`, decreasing $\alpha_0$ toward 0 (full AR) yields validation perplexities that **closely match those of AR models**.
>
> **New Experiments:** As a sanity check, we retrained Eso-LMs with $\alpha_0=0$ on LM1B and OWT, recovering exact AR perplexities of `22.38` and `17.90`, respectively.
>
> **Advantage: Eso-LMs provide a speed–quality tradeoff unavailable to AR models**. As shown in `Figs. 3, 4`, they achieve comparable sample quality to AR under similar sampling-time budgets while offering much faster generation.

---

> ### Author Response · Authors · 2025-11-21
> **Response (5/6) to Reviewer RnKE**
>
> # Concern 5: Downstream performance
>
> > The combination of AR and diffusion objectives feels somewhat mechanical—focused on unifying two training losses rather than addressing a specific modeling limitation or data property.
> >
>
> We respectfully disagree. **The combination of AR and diffusion objectives is** **not mechanical**; **varying $\alpha_0$ for Eso-LMs enables a principled tradeoff between speed and quality** that neither pure diffusion nor pure autoregression can achieve.
>
> Pure diffusion is suboptimal because, even when the number of sampling steps matches that of AR models, the generated sample quality remains lower. **The $\alpha_0$ hyperparameter provides a systematic way to integrate autoregression into the generation process** which ensures that  increasing the sampling budget significantly boosts sample quality compared to using $\alpha^\text{train/eval}_0 = 1$ alone.
>
> **Downstream performance on unconditional text generation** (taken from `Figs. 3, 4`):
>
> |  | Best Diffusion Only ($\alpha^\text{train/eval}_0=1$) | Best across multiple $\alpha^\text{train/eval}_0$ |
> | --- | --- | --- |
> | Gen PPL ($\downarrow$) | 48.9 | **23.2** |
> | Mauve ($\uparrow$) | 0.54 | **0.83** |
>
> **Takeaway:** Different $\alpha_0$ values are optimal for different sampling-time budgets, demonstrating that controlled AR–Diffusion interpolation yields a superior speed–quality tradeoff.
>
> > It remains unclear what type of data/scenario actually benefits from this interpolation, e.g., general, coding, math, agent, etc.
> >
>
> Training models that perform well on the above mentioned downstream tasks requires large scale training. As a small academic lab, **we chose to allocate our compute budget to ablate every individual component of our method** where we trained `34` models amounting for almost `9000` H200 GPU hours as outlined in our earlier response.

---

> ### Author Response · Authors · 2025-11-21
> **Response (6 / 6) to Reviewer RnKE: Comments**
>
> ## Comment 1: Accommodating multiple $\alpha_0$ during training
>
> > If my understanding is correct, for different values of \alpha_0, we have to re-train the model from scratch.
> >
>
> Yes, that is correct.
>
> > Is there some potential methods that can accommodate multiple \alpha_0's in one training run?
> >
>
> One possibility is to train the model with a variable $\alpha^\text{train}_0 \sim \mathcal{U}[0, 1]$ during training. We leave this exploration for future work.
>
> In our paper, we show empirically that training with $\alpha^\text{train}_0 = 1$ (full diffusion) and adjusting $\alpha^\text{eval}_0$ during inference **is generally sufficient for effective AR–Diffusion interpolation** `[Lines 473-475]`. As shown in `Fig. 16`, **tuning $\alpha^\text{train}_0$ can yield a better performance but is not necessary.**
>
> ---
>
> ## Comment 2: Reason for divergence for certain hyperparameter values in ablation
>
> As noted in `Lines [205–215]`, the **hybrid training objective** (Eq. 7) combines an AR loss and an MDM loss, each with different variance characteristics [3]. The parameter $\kappa$ determines the fraction of the batch trained with the diffusion loss.
>
> A small $\kappa$ means fewer samples contribute to the diffusion objective, amplifying its already high variance. Prior work [3, 4] shows that diffusion training is degraded with a smaller batch size. Through extensive ablations (`Tab. 4`), we find that **$\kappa = 0.5$ offers the most stable training** across various $\alpha^\text{train}_0 \in \\{0.0625, 0.125, 0.25, 0.5\\}$ values.
>
> ---
>
> ## Comment 3: Comparison with any-order GPT
>
> > How does this work relate to the any-order GPT line of work [1,2], which also kind of combines GPT and MDM?
> >
>
> We thank the reviewer for pointing us to these papers. **Any-order GPT** [2] is a concurrent work, and both **$\sigma$-GPT** [1] and **Any-order GPT are a special case of Eso-LMs when** $\alpha_0 = 1$.
>
> **Advantages of Eso-LMs over [1, 2]:**
>
> 1. **Interpolation ability:** Eso-LMs interpolate between AR and diffusion in terms of sample quality, while [1, 2] do not. Their behavior is closer to MDLM.
> 2. **Architectural simplicity:** [2] requires substantial architectural modifications to a decoder-only transformer.
>     1. They shuffle the clean sentence $\mathbf{x}$ according to an ordering $\sigma$ and train the model to predict $\mathbf{x}\_{\sigma(i+1)}$ *from $\mathbf{x}_{\sigma(i)}$*. Empirically, this works only when each token is associated with positional embeddings of both $\sigma(i)$ and $\sigma(i+1)$. To achieve this, the authors modify the architecture of the causal transformer and introduce target positions via **Adaptive LayerNorm**, which adds learnable parameters.
>     2. In contrast, **Eso-LMs introduce no additional parameters**. We achieve the same objective by designing **custom attention masks**, implemented in just a few lines of code (Lines [307–310, 341–343]).
>
> ---
>
> **References**
>
> [1] σ-GPTs: A New Approach to Autoregressive Models, arxiv 2404.09562
>
> [2] Any-Order GPT as Masked Diffusion Model: Decoupling Formulation and Architecture, arxiv 2506.19935
>
> [3] Sahoo et al., 2024 "Simple and effective masked diffusion language models."
>
> [4] Kingma et al., 2021 "Variational Diffusion Models."

---

### Official Review · Reviewer_stDQ · 2025-11-02

**Soundness:** 3
**Presentation:** 3
**Contribution:** 2
**Rating:** 4
**Confidence:** 4

**Summary:**

This paper proposes Esoteric Language Models (Eso-LMs), a new model family that fuses autoregressive (AR) and Masked Diffusion (MDM) models, aiming to get the best of both: the quality and efficiency of AR models and the parallel generation capabilities of MDMs. Specifically, it introduces a novel training procedure and attention mechanism that enables KV caching during the diffusion phase. This is achieved by replacing the standard bidirectional attention in MDMs with a causal attention mechanism on a shuffled sequence, allowing a single, unified KV cache to be shared across both parallel (diffusion) and sequential (AR) generation phases. The model is trained using a hybrid objective that combines a standard autoregressive (AR) loss with a Masked Diffusion Model (MDM) loss , allowing the model to interpolate between the two paradigms. The experiments show that Eso-LMs achieve a new state-of-the-art on the speed-quality Pareto frontier among MDMs, and on long contexts, they are 14-65x faster than standard MDMs and 3-4x faster than prior semi-autoregressive approaches.

**Strengths:**

1. KV caching is a key problem in MDM generation. Eso-LMs achieve massive inference speedups, especially on long contexts.
2. The authors identify key shortcomings in the previous hybrid model, BD3-LM, such as "degraded samples at low sampling steps" and "incomplete caching". Eso-LMs are shown to overcome these limitations and outperform BD3-LMs. Especially, Eso-LMs remain competitive with MDMs in the low-NFE regime and with AR models in the high-NFE regime.
3. The model uses a new training and sampling procedure that replaces bidirectional attention with causal attention on a shuffled sequence. This is the key innovation that enables KV caching during the diffusion phase.
4. Computing importance-weighted bounds for discrete diffusion models is also an interesting contribution.

**Weaknesses:**

1. Perplexity still lags behind AR models. The hybrid training objective pushes the model closer to AR models, but there is still a gap (24.51 vs 22.38 on LM1B, 20.86 vs 17.90 on OWT).
2. Replacing bidirectional attention with causal attention to enable KV caching comes at a cost. When Eso-LM is trained as a pure diffusion model ($\alpha_0=1$), its perplexity is noticeably worse than the standard MDLM baseline (which uses bidirectional attention).
3. When evaluated on unseen datasets (zero-shot likelihood evaluation in Appx E2), the Eso-LMs performed consistently worse than all other baselines, including the standard AR model, MDLM, and even BD3-LMs.

**Questions:**

Why is the zero-shot generalization worse than baselines? Is such overfitting expected? What are the possible reason for this?

Why is the training time with $\alpha_0<1$ 1.37x of the pure MDLM ($\alpha_0=1$)?

---

> ### Author Response · Authors · 2025-11-18
> **Response (1/4) to Reviewer stDQ**
>
> We appreciate the reviewer’s recognition that our method is novel and addresses two key problems in MDM generation (KV caching for MDM, hybrid AR-MDM with shared KV cache) to achieve a greatly improved speed-quality trade-off over existing methods.
>
> We now address the reviewer’s comments and questions below.
>
> # Concern 1: Perplexity Gap between AR and Eso-LMs
>
> The $\alpha_0$  hyperparameter in the hybrid training objective enables interpolation between AR and diffusion models, where $\alpha_0=0$ corresponds to a fully AR model and  $\alpha_0=1$ to a fully diffusion model (Sec 3.1, last paragraph).
>
> > The hybrid training objective pushes the model closer to AR models, but there is still a gap (24.51 vs 22.38 on LM1B, 20.86 vs 17.90 on OWT).
>
> These validation PPLs of Eso-LMs on LM1B ( `24.51`) and OWT (`20.86`) correspond to $\alpha_0 = 0.0625$ and  $\alpha_0=0.125$ respectively— values that are **not fully AR**. Hence, **the observed gaps are expected**.
>
> **New Experiments**: As a sanity check, we retrained our method, Eso-LMs, on LM1B and OWT with $\alpha_0 = 0$, and demonstrate that **Eso-LMs $\alpha_0 = 0$ match AR PPLs of `22.38` and `17.90`**.

---

> ### Author Response · Authors · 2025-11-18
> **Response (2/4) to Reviewer stDQ**
>
> # Concern 2: Eso-LMs worse than MDLM at $\alpha_0=1$
>
> > Replacing bidirectional attention with causal attention to enable KV caching comes at a cost. When Eso-LM is trained as a pure diffusion model ($\alpha_0=1$), its perplexity is noticeably worse than the standard MDLM baseline (which uses bidirectional attention).
>
> **As acknowledged in `[Lines 422-424]`, this is expected.** Mathematically, perplexity of a diffusion model represents sample quality under $\infty$ sampling budget and doesn’t capture performance in realistic settings with finite sampling time. Consequently, **this metric fails to reflect the** **practical advantages** that Eso-LMs provide over MDLM [1].
>
> **Advantage 1: Eso-LMs outperform MDLM on the speed-quality Pareto frontier and establish a new SOTA**
>
> 1. Due to sparse attention, every denoising step in Eso-LMs significantly faster than MDLM.
> 2. Therefore, under a fixed sampling-time budget, Eso-LMs can perform **more denoising steps** and produce **higher-quality samples**, as shown in `Figs. 3, 4`.
>
> **Advantage 2: Eso-LMs support exact likelihood estimation unlike MDLM**
>
> 1. The paper introduces, for the first time, a formula to compute the exact likelihood for Masked Diffusion Models, applicable to both MDLM and Eso-LMs:
> $-\log p(x) = \lim_{K \to \infty} - E_{\sigma_1,\ldots,\sigma_K} \left[ \log \frac{1}{K} + \log \sum_{k=1}^K \exp\left({\sum_{l=1}^L \log p_\theta (x_{\sigma_k(l)} \mid x_{\sigma_k(<l)})}\right) \right]$ where $\sigma_k$ denotes a permutation of the clean sentence $x$.
>
> 2. In practice, $K \approx 100$ suffices to approximate the exact likelihood.
> 3. However, as described in `[Lines 261-264]`, **the formula is intractable for MDLM,** since  evaluating the quantity ${\sum_{l=1}^L \log p_\theta (x_{\sigma_k(l)} \mid x_{\sigma_k(<l)})}$ requires $L$ (typically $L \approx 1000$) forward passes. In contrast, Eso-LMs can compute this in a single forward pass.
>
> ---
>
> **Reference**
>
> [1] Sahoo et al., 2024 "Simple and effective masked diffusion language models."

---

> ### Author Response · Authors · 2025-11-18
> **Response (3/4) to Reviewer stDQ**
>
> # Concern 3: Poor zero-shot likelihood
>
> **Limitations of likelihood-based metrics:** As noted in our earlier response to Concern 2, likelihood-based metrics such as perplexity and zero-shot perplexity **do not account for inference speed**, and therefore are **not ideal for comparing** Eso-LMs with baselines such as MDLM [1] and BD3-LMs [2].
>
> > When evaluated on unseen datasets (zero-shot likelihood evaluation in Appx E2), the **Eso-LMs performed consistently worse than all other baselines**, including the standard AR model, MDLM, and even BD3-LMs.
> >
>
> **This statement is not quite accurate.** The zero-shot results for MDLM and BD3-LMs in Table 5 correspond to publicly released checkpoints trained for 1 M steps, whereas Eso-LMs were trained for only **250 K steps**. We re-evaluated all models using 250 K-step checkpoints for a fair comparison, yielding the updated results below:
>
> |  | OWT Val. | PTB* | Wikitext* | LM1B* | Lambada | AG News* | Pubmed | Arxiv |
> | --- | --- | --- | --- | --- | --- | --- | --- | --- |
> | AR **(new**) | 17.90 | 82.00 | 26.54 | 52.14 | 51.69 | 55.53 | 49.49 | 44.98 |
> | Eso-LMs ($\alpha_0=0.125$)  | 21.87 | 97.46  | 35.65  | 60.11 | 69.13 | 65.26 | 65.27 | 57.4 |
> | MDLM **(new)** | 25.19 | 100.17 | 37.08 | 70.79 | 52.06 | 71.37 | 46.51 | 40.21 |
> | BD3-LMs ($L’=16$) **(new)** | 23.57 | 95.87 | 32.88 | 65.11 | 50.05 | 61.68 | 43.41 | 40.13 |
>
> `Key Takeaways`:
>
> 1. All models were trained on OpenWebText (OWT). On **4 out of 7** (PTB, Wikitext, LM1B, AG News) unseen datasets, the **ordering of AR < Eso-LMs < MDLM** is preserved, **consistent with the OWT validation results**.
> 2. On **4 of 7** datasets (PTB, Wikitext, LM1B, AG News), **Eso-LMs perform on par with BD3-LMs ($L’=16$)**.
>
> **Note:** Importantly, when both **sampling speed** and **sample quality** are considered, **Eso-LMs establish a new state-of-the-art on the speed–quality Pareto frontier**, substantially outperforming MDLM and BD3-LMs, as demonstrated in `Figs. 3 and 4`.
>
> ---
>
> **References**
>
> [1] Sahoo et al., 2024 "Simple and effective masked diffusion language models."
>
> [2] Arriola et al., 2025 "Block Diffusion: Interpolating Between Autoregressive and Diffusion Language Models."

---

> ### Author Response · Authors · 2025-11-18
> **Response (4/4) to Reviewer stDQ**
>
> # Clarification 1: Training time difference for $\alpha_0<1$ vs $\alpha_0=1$
>
> > Why is the training time with $\alpha_0<1$ 1.37x of the pure MDLM ($\alpha_0=1$)?
>
> As noted in `[Lines 205–215]`, the hybrid training objective (Eq. 7) consists of two components: an AR loss and an MDM loss.
>
> When $\alpha_0 = 1$, the AR loss vanishes, and the entire batch is used for the MDM loss and the training time matches MDLM’s [1].
>
> When $0 < \alpha_0 < 1$,
> * the model is trained jointly with both losses, each computed on **half of the batch**.
> * For the AR loss, we **concatenate the clean sequence with its partially masked version**, effectively **doubling the input length** (see `[Fig. 2, left]`).
> * As noted in `Sec. 4.1.2`, since only half the batch uses doubled-length inputs, overall training is `1.37×` slower than pure MDLM.
>
> **Speed-up relative to BD3-LMs** [2]: Our main baseline, BD3-LMs, also interpolates between AR and diffusion. However, it **doubles the sequence length for the entire batch**, making it **2.67× slower than MDLM**. In contrast, **Eso-LMs are ≈2× faster to train than BD3-LMs** while outperforming them on speed-quality Pareto frontier `Figs. 3, 4`.
>
> ---
>
> **References**
>
> [1] Sahoo et al., 2024 "Simple and effective masked diffusion language models."
>
> [2] Arriola et al., 2025 "Block Diffusion: Interpolating Between Autoregressive and Diffusion Language Models."

---

### Author Response · Authors · 2025-12-01
**General Response**

# Current Diffusion LM Paradigms and Their Limitations

The field of language modeling is beginning to see a notable shift as major industry players such as Google (`Gemini Diffusion`), ByteDance (`Seed Diffusion` [1]), and Inception Labs (`Mercury Coder` [2]) have started to invest in diffusion-based language models as alternatives to traditional LLMs due to their promising speed–quality tradeoffs. Notably, these models [1, 2] are built on the MDLM [3] and BD3-LMs [4] frameworks.

In this paper, we identify three critical limitations in these approaches:

1. **MDLM lacks support for KV caching**, hindering its efficiency in generation.
2. **BD3-LMs** support partial KV caching, **but tend to produce degenerate outputs** when the number of sampling steps is reduced—limiting their usability in low-latency settings.
3. **They both lack tractable likelihood estimation.** Given a single sequence:
    1. Likelihood estimation requires a large number of forward passes, rendering policy-gradient style RL computationally intractable and necessitating approximations [5].
    2. Exact likelihood estimation is computationally intractable.

# Our Contributions

We **propose a new hybrid framework for language modeling** called Esoteric Language Models (Eso-LMs) that **interpolate AR and MDM paradigms** and tackle the aforementioned limitations. Our main contributions are:

1. Eso-LMs achieve **fine-grained interpolation between AR and MDM perplexities**, narrowing the gap to AR models `Sec. 5.1`.
2. By enabling KV caching during diffusion while preserving parallel generation, **Eso-LMs achieve a new state of the art on the speed-quality Pareto frontier** for unconditional generation `Sec 5.2`. At longer contexts, our method achieves 14−65× faster inference than standard MDMs and 3−4× faster inference than BD3-LMs `Sec 5.3`.
3. Using Eso-LMs, **for the first time, we enable exact likelihood compution for masked diffusion models**  `Sec. 5.1`.

---

## References

[1] Seed et al., 2025 “Seed Diffusion: A Large-Scale Diffusion Language Model with High-Speed Inference”.

[2] Khanna et al., 2025 “Mercury: Ultra-Fast Language Models Based on Diffusion”.

[3] Sahoo et al., NeurIPS 2024 "Simple and Effective Masked Diffusion Language Models".

[4] Arriola et al., ICLR 2025 "Block Diffusion: Interpolating Between Autoregressive and Diffusion Language Models".

[5] Zhao et al., NeurIPS 2025 “d1: Scaling Reasoning in Diffusion Large Language Models via Reinforcement Learning”.

[6] Sahoo et al., ICML 2025 “The Diffusion Duality”.

---

> ### Author Response · Authors · 2025-12-01
> **A summary of our rebuttal**
>
> The reviewers affirm the relevance and significance of our contributions while offering the following comments for further clarification and improvement.
>
> Notation: In our work, $\alpha_0$ represents the proportion of tokens modeled or generated by masked diffusion.
>
> # Reviewer **stDQ**
>
> Reviewer stDQ is mainly concerned about **whether our proposed Eso-LMs actually interpolate between AR and MDLM perplexities**.
>
> We emphasized that **this interpolation is clearly shown** **in** `Table 1 & 2` : in terms of PPL, Eso-LM is **comparable to AR** when $\alpha_0$ is close to 0, worse than MDLM when $\alpha_0=1$, and **comparable to MDLM** for some intermediate $\alpha_0$ value. During rebuttal, we added a sanity-check experiment for $\alpha_0=0$, at which Eso-LM recovers the exact AR PPL as expected.
>
> ---
>
> Reviewer stDQ further inquired why Eso-LM with $\alpha_0=1$ has worse PPL than MDLM.
>
> We noted that **this result is expected** when switching from bidirectional to sparsified causal attention to enable KV caching, **as mentioned in our submission**. Also, PPL represents sample quality under $\infty$ sampling budget, which **fails to represent the practical advantages of Eso-LMs**:
>
> 1. Eso-LMs outperform MDLM on the speed-quality Pareto frontier and establish a new SOTA;
> 2. Eso-LMs support exact likelihood estimation unlike MDLM.
>
> We also provided clarifications to questions regarding zero-shot likelihood and training time.
>
> # Reviewer **RnKE**
>
> Reviewer RnKE is mainly concerned about the complexity of our method.
>
> We clarified that **the modifications introduced by Eso-LMs relative to MDLMs are minimal**, and can be implemented in only a dozen lines of code, which we have added to our Appendix (Page 22, Fig. 11) during rebuttal.
>
> ---
>
> We emphasized that the combination of AR and diffusion objectives is not purely mechanical: varying the proportion ($\alpha_0$) of each with **Eso-LMs enables a principled tradeoff between speed and quality** on the task of unconditional text generation.
>
> Regarding concerns about insufficient justification for technical details and limited ablations, we
>
> - Cited the relevant sections of our initial submission for justification;
> - **Enumerated all ablation studies performed for every aspect of our method, which adds up to** `9000 H200 GPU hours`.
>
> # Reviewer **hFRL**
>
> Reviewer hFRL is mainly concerned about three aspects: (1) why Eso-LMs can obtain a better speed-quality trade-off than baselines, (2) whether exact likelihood evaluation of Eso-LMs via the importance-weighted (IW) bound is a valuable contribution, and (3) performance on down-stream tasks.
>
> To explain the speed-quality tradeoff, we elaborated on two innovations made by Eso-LMs:
>
> 1. Enabling KV caching during the diffusion phase so sampling is faster or on par with AR;
> 2. Providing a principled method to transition toward AR generation (controlled by $\alpha_0$) as the sampling time budget increases, which bridges the qualitative gap between AR and diffusion-based generation.
>
> Together, **these innovations place Eso-LMs ahead of baselines on the speed–quality Pareto frontier and establish a new SOTA** `Figs. 3 & 4` **.**
>
> ---
>
> Likelihood / perplexity are key metrics for evaluating LLMs. For diffusion models, however, only a likelihood bound **is typically available, which makes perplexity-based comparisons unfair**. To address this:
>
> - We introduce **the first exact likelihood formula for discrete diffusion models** using IW bounds.
> - Identify that **Eso-LMs are the first diffusion models that can evaluate this bound efficiently.**
>
> For downstream tasks, the most meaningful metrics are the ones that capture both sample quality and sampling speed. Therefore, we focus on the speed–quality tradeoff in `Figs. 3 and 4`, which are the most relevant evaluations. While coding and reasoning are important future directions, they are beyond our computational capacity as a small academic lab.
>
> ---
>
> ## References
>
> [1] Seed et al., 2025 “Seed Diffusion: A Large-Scale Diffusion Language Model with High-Speed Inference”.
>
> [2] Khanna et al., 2025 “Mercury: Ultra-Fast Language Models Based on Diffusion”.
>
> [3] Sahoo et al., NeurIPS 2024 "Simple and Effective Masked Diffusion Language Models".
>
> [4] Arriola et al., ICLR 2025 "Block Diffusion: Interpolating Between Autoregressive and Diffusion Language Models".
>
> [5] Zhao et al., NeurIPS 2025 “d1: Scaling Reasoning in Diffusion Large Language Models via Reinforcement Learning”.
>
> [6] Sahoo et al., ICML 2025 “The Diffusion Duality”.

---

### Meta-Review · Area_Chair_Kg3V · 2026-01-08

**Summary:**

This paper proposes a hybrid model that interpolates between the AR and MDM modes. It works by modeling and generating a subset of tokens with MDM, then handles the remaining ones the AR. Another contribution is enabling KV cache for the MDM mode but enforcing a causal Transformer architecture. Experiments demonstrate considerable speedups compared to MDM baselines.

This appears to be a sound and timely idea, as acknowledged by most reviewers. After going through the reviews and rebuttal, one major outstanding concern I can see is the ones raised by reviewer stDQ which I also resonate with. In other words, the hybrid model is supposed to be striking a good tradeoff between the AR and MDM model, but it's not entirely clear how it's better than using AR alone (which is certainly not the case judging from PPL). I'd encourage authors to further flesh out the narrative and evaluations to better position this work -- which I think is very promising but just needs more work. Based on these reasons, I regrettably recommend reject.

**Reviewer Concerns:**

Reviewer stDQ's concerns don't seem fully addressed.

**Reviewer Scores:**

Reviewer stDQ will probably maintain a negative score

---

### Decision · Program_Chairs · 2026-01-26

Reject